# Spatiotemporal heterogeneity of bicycle ridership based on GTWR model

**Xiaonan Zhang**[1¤*], **Xiaohui Yan**[2], **Borui Yan**[1], **Shuaiyang Jiao**[3], **Lei Zhang**[3]

**1** Xianyang Normal University, Xian Yang, Shaanxi, China, **2** Shanghai Urban Construction Vocational College, Shanghai, China, **3** Henan University of Urban Construction, Pingdingshan, China

¤ Current address: Xianyang Normal University, Wenlin Road, Xian Yang, Shaanxi Province, China
* 2018021070@chd.edu.cn

## Abstract

As a low-carbon, green and environmentally friendly mode of travel, bicycles possess significant advantages in short-distance trips. In previous studies on the relationship between the built environment and bicycle behavior, the built environment variable only took into account the number or density of facilities. However, due to their different grades and formats, the attractions of similar facilities of the same size to residents vary considerably. Therefore, this paper constructs a comprehensive index of POI (Point of Interest) facility quality to reflect the influence of the number of facilities and preferences on bicycle trips. In addition, two types of riding safety indicators, namely the proportion of non-isolation bars and the proportion of non-motor vehicle lane parking, are added to the road safety facilities. On this basis, GWR and GTWR models are established to explored the temporal and spatial distribution characteristic of cycling, and identifies the relationship between cycling behavior and built environments based on 2022 Daily Trip Survey in Xianyang, China. The model results demonstrate the following: (1) The GTWR model exhibits a better fit compared to the GWR model. (2) There are significant differences between the urban central area and the marginal area, which verifies that similar facilities have diverse impacts on the cycling frequency in distinct regions. (3) The promoting or inhibiting effects of the urban built environment on the cycling frequency are highly congruent with the temporal characteristics of commuting, and these effects typically reach their maximum during commuting rush hours. (4) Cycling safety facilities constitute a significant factor influencing the cycling frequency. These results can not only offer guidance for urban planning and design but also foster the sustainable development of green transportation.

## 1. Introduction

The escalating number of motor vehicles presents significant challenges to traffic efficiency in small and medium-sized cities. Currently, due to the rapid expansion of car travel, traffic congestion and environmental pollution are becoming increasingly severe. The shift in travel patterns from private vehicles to non-motorized transport has emerged as a prominent topic in the domain of urban transportation management. With the merits of flexibility, convenience, low carbon emissions, and health benefits, non-motor vehicle travel constitutes a

**Data availability statement:** All relevant data are within the paper and its Supporting information files.

**Funding:** The author(s) received no specific funding for this work.

**Competing interests:** The authors have declared that no competing interests exist.

substantial proportion in people's short-distance trips and serves as an essential travel mode for urban residents, especially in small and medium-sized cities.

Previous research on cycling behavior predominantly centered on the influencing factors and cycling characteristics. The principal influencing factors of residents' travel behavior encompass socio-economic attributes, such as population, gender, income, occupation, vehicle ownership status, residential type, as well as trip attributes, including travel time, distance, and purpose. Additionally, certain studies have indicated that the built environment is a crucial determinant of non-motor travels [1,2]. Land use diversity, density, and street network characteristics are commonly regarded as the primary aspects of the built environment [3,4]. A cross-sectional study utilizing data from 10 countries disclosed that some objectively measured building environmental attributes, like land use mix, street connectivity, and residential density, were associated with cycling [5]. A study carried out in the UK revealed that perceived street connectivity was correlated with cycling [6]. A longitudinal study in Australia demonstrated that cycling trips were found to enhance street connectivity and destination accessibility. Mixed land use and commercial street connectivity have a favorable effect on the willingness to walk and cycle. According to the investigation results in San Francisco, factors such as land use, density, and street connectivity would promote walking and bicycle trips when the travel distance is less than 5 miles [7]. Moreover, the influence of locations, housing prices, distance from the central business district (CBD), etc. were all considered as built environment indicators [8,9]. Besides, bicycle infrastructure, high intersection density, and diversified land use are positively correlated with non-motorized travel [5,10]. Urban environments equipped with dedicated bicycle infrastructure, traffic mitigation measures, and medium or high urban density are associated with higher bicycle usage rates. Cervero et al. explored the relationship between microscale factors or environmental aesthetics, such as greenery and trees, benches and street lights, pleasant buildings and landscapes, and cycling frequencies [7].

The land use characteristic can be acquired through GIS (geographical information system). Thermal point data and POI (point of interest) data can be obtained from Baidu or Gaode maps, and the street network data can be retrieved via the Open Street Map. The urban built-up environment refers to the interactive space environment composed of land use, traffic infrastructure, urban design, and other elements, which can provide space for human activities [11]. The concept of the built-up environment was first introduced by Cervero [7], comprising density, diversity, and design. When the accessibility to the destination and the distance from the transportation facilities are augmented, the built-up environmental characteristics are extended to 5D [11]. Regarding the bicycle cycling characteristics, the research mainly focused on travel demand intensity, trip purpose, travel mode, travel distance, and so forth. Vogel et al. conducted a study on the demand characteristics of public bicycles during peak travel periods in Lyon on both weekdays and weekends [12]. Guo performed a visual comparative analysis of the OD (origin-destination) intensity of public bicycle stations for various purposes (commuting and living), which are situated at different types of facilities (residence, administrative office, campus, road, subway entrance, etc.), and investigated the route preference based on the shortest path theory [13]. Suo adopted the equally spaced strength grading method to divide the number of shared bicycles and parking intensity into 6 levels and carried out a detailed analysis of each level separately [14]. Zhou et al. analyzed the characteristics of hotspot areas, riding length, turnover, and the work-residence balance based on Mobike cycling data in Wuhan, Xi'an, and Hangzhou [15]. Mo et al. explored the temporal and spatial distribution characteristics of shared bicycle trips OD in Guangzhou's old city based on Mobike cycling data and analyzed the influencing factors during peak hours [16]. Zhang et al. established a multiple linear regression model to validate the impact of the built environment on travel [17]. Ahmadreza et al. employed a multilevel statistical

regression model to examine the impact of meteorological, temporal, infrastructure, land use, and built environment attributes on passengers' arriving and leaving flows at the stations [18]. Ahmadreza et al. and Cui et al. studied the impact of spatial and temporal effects on the bike-sharing system [19,20].

In summary, the extant research concerning the impact of the built environment on travel behavior chiefly manifests several limitations. Firstly, the majority of studies resort to global modeling techniques, centering on socio-economic attributes and travel characteristics, while frequently disregarding the spatio-temporal heterogeneity of the built environment's impacts. Although a handful of studies have endeavored to probe into this heterogeneity, they are preponderantly concentrated in developed countries, thereby leaving a conspicuous lacuna in research for developing nations such as China. Secondly, in the examination of the built environment, contemporary research has a propensity to accentuate the quantity or type of facilities, overlooking pivotal factors like their quality, grade, and attractiveness. For example, supermarkets and stores of identical size might differ substantially in attractiveness; likewise, schools of disparate grades have diverse influences on travel behavior, especially in China where higher-quality schools customarily possess stronger allure. Moreover, the significance of cycling safety facilities in fostering cycling behavior is often overlooked, notwithstanding their essential role in augmenting cycling safety and inducing more individuals to opt for cycling.

In light of these research gaps, this study takes the weekday cycling data from Xianyang, Shaanxi Province, China as an instance, and utilizes geospatial analysis models to comprehensively contemplate the safety of road infrastructure, the quality of daily service facilities, and cycling safety facilities. It delves into the influence of assorted factors on the spatio-temporal characteristics of cycling. This study aims to bridge the research void in spatio-temporal heterogeneity studies within developing countries, while underscoring the importance of built environment quality and cycling safety facilities. The outcomes furnish a scientific foundation for urban planning or policy formulation, with the objective of promoting environmentally friendly travel, diminishing transportation energy consumption and carbon emissions, and steering cities towards a more salubrious and sustainable development trajectory.

This paper is structured as follows. The subsequent section acquaints the methodology of this study. Section three elucidates the data sources and the variable selection. Section four recapitulates the spatio-temporal characteristics of the principal factors and proffers further elucidation. The results are deliberated and policy recommendations are proffered in the final section.

## 2. Spatial analysis methodology

The mainstream approaches for spatial characteristic analysis encompass the GWR (geographically weighted regression) model, SLM (spatial lag model), SEM (spatial error model), SAM (spatial auto-correlated model), SDM (spatial Durbin model), and SIM (spatial interaction model). Wu et al. utilized both a global regression model and a GWR model to investigate the global and local impacts of the built environment on bicycle usage [21]. Cheng et al. introduced a multi-scale geographically weighted regression (MGWR) model and took into account six types of "D" variables, namely density, diversity, design, destination accessibility, transportation distance, and demand management [22]. Liu et al. proposed a comparative method of hierarchical clustering and multinomial logit regression to identify the spatio-temporal patterns of You-Bike in the Taipei Metropolitan Area [23]. Yang et al. employed a semi-parametric geographically weighted regression (S-GWR) model to explore the spatial relationships among the predicting variables, response variables, and the amount of bicycle riding [24]. Ji et al. studied the public and bicycle modes respectively and disclosed the

differences in the factors for different bicycle modes [25]. Ma et al. incorporated time factors to establish a Geographical and Temporal Weighted Regression (GTWR) model and revealed the influence of land use and sociodemographic attributes on public bicycle demand [26].

In addition to the aforementioned common linear regression models, non-linear models have been widely adopted to study travel behavior, such as the machine learning methods Random Forest (RF), Support Vector Machine (SVM), BOOST, and Artificial Neural Network (ANN). Hagenauer and Cheng both discovered that RF exhibits high accuracy and can be employed for travel mode choice prediction [27,28]. Tao constructed a Gradient Boosting Decision Tree (GBDT) model to study the significance of the explanatory variables and the spatial dependence, and the results verified the non-linear relationship between the spatial characteristic and travel distance [29]. Chen introduced the Gradient Boosted Regression Trees (GBRT) to analyze the effective range and threshold effect of the non-linear impact of the built environment on the travel behavior of shared bicycles through a partial dependency graph [30]. Luan proposed a negative binomial-based additive decision tree (NBADT) model to study the non-linearity of bicycle reverse riding behavior and built-up environmental factors and to analyze the perception of the factors contributing to these hazardous behaviors [31].

MGWR, being an enhanced geographical weighted regression approach, augments the performance of the GWR model by allotting a separate bandwidth to each predictor variable. Nevertheless, in our research, the emphasis is placed on the spatiotemporal heterogeneity of the influence of the built environment on travel behavior. Although MGWR is proficient in dissecting spatial heterogeneity, it regrettably does not take into consideration the unequal temporal distribution of sample points, thereby restricting its suitability within our research framework. Analogously, the SVCM model, which is another potent spatial varying coefficient model, is also incapable of directly portraying the temporal heterogeneity of sample points. Our research endeavors to explore the impact of the built environment on travel behavior from both spatial and temporal perspectives, rendering it highly crucial to select a model that can capture both spatiotemporal heterogeneities.

Existing studies have verified the existence of an unstable state between bicycle travel characteristics and the built environment [32,33], which aligns well with the requisites of the GWR model and the GTWR model concerning the spatial heterogeneity of data. Moreover, as the travel distance elongates, the likelihood of opting for bicycle travel diminishes, which concurs with the fundamental principle that the observed values nearer to the distance location point exert a lesser impact on the dependent variable at the estimation point. Hence, this study opts for the GWR model and the GTWR model respectively to probe into the spatial non-stationarity of cycling frequency.

## 2.1. GWR model and GTWR model

Geographic weighted regression (GWR) represents one of the principal methods for investigating the spatial disparities of factors [21,22]. In the context of the GWR model, the geographical locations of data have been incorporated into the regression parameters, subsequently enabling the exploration of the quantitative relationships among certain variables possessing spatial distribution characteristics. Consequently, the GWR model is capable of illustrating the spatial influence of structural alterations in environmental variables within built-up areas on the distribution of bicycle ridership. The general form of the GWR model is presented as Eq. (1).

$$y_i = \beta_0\left(u_i, v_i\right) + \sum_{k=1}^{p} \beta_{ik}\left(u_i, v_i\right)x_{ik} + \varepsilon_i \left(i = 1, 2, \ldots\ldots n\right) \tag{1}$$

Where $y_i$ represents the dependent variable, that is the frequency of cycling. $x_{ik}$ represents the kth independent variable of sample $i$. Each sample means a cycling trip, and $x_{ik}$ is just the influence fator. $u_i$ represents the longitude coordinate of the $i$th sample. $v_i$ represents the latitude coordinate of the $i$th sample, $(u_i, v_i)$ is the geographical location factor of sample $i$. $\beta_{ik}(u_i, v_i)$ is the kth regression parameter of sample i and the geographical location function. $\varepsilon_i \sim N(0, \delta^2)$, $Cov(\varepsilon_i, \varepsilon_i) = 0 (i \neq j)$. It can be abbreviated as Eq. (2):

$$y_i = \beta_{i0} + \sum_{k=1}^{p} \beta_{ik} x_{ik} + \varepsilon_i \, (i = 1, \, 2, \, \ldots\ldots n) \tag{2}$$

When $\beta_{1k} = \beta_{2k} = \ldots = \beta_{nk}$, the above model can be converted into a linear regression model.

The Geographical and Temporal Weighted Regression (GTWR) model incorporates time data into regression parameters, taking the time dimension into account. In this way, the spatial and temporal variations of each parameter can be measured simultaneously [26]. The structure of the GTWR model is presented as Eq. (3):

$$Y_i = \beta_0(u_i, v_i, t_i) + \sum_{k=1}^{p} \beta_{ik}(u_i, v_i, t_i) x_{ik} + \varepsilon_i \, (i = 1, \, 2, \, \ldots\ldots n) \tag{3}$$

Where, $Y_i$ represents the dependent variable of the ith sample, that is the frequency of cycling. $(u_i, v_i, t_i)$ is the geographical location factor of sample $i$, that is the geographical and temporal dimension coordinate. $\beta_0(u_i, v_i, t_i)$ is the constant term. $\beta_{ik}(u_i, v_i, t_i)$ is the kth regression parameter of sample i. Other variables have been defined the same as the GWR model.

The essence of the GWR and GTWR model regression is that data is closer has a greater impact on the results in contrast to data that is farther away. The attenuation function is a crucial index for calculating the different spatial positions of each factor in spatial weighting. The attenuation function assigns different weights to samples based on their distinct spatial relationships. Subsequently, the regression coefficients for each sample can be derived. The attenuation function is presented as Eq. (4):

$$\breve{\beta}(u_i, v_i, t_i) = \left[ X^T W(u_i, v_i, t_i) X \right]^{-1} X^T W(u_i, v_{i,t_i}) Y \tag{4}$$

Where, $W(u_i, v_i, t_i)$ is the spatial-temporal weight matrix (that is the distance matrix). The weight matrix is the important basis to carry on separate regression analysis for different spatial locations.

## 2.2. Spatiotemporal weight function

The spatial-temporal weight function exerts a direct influence on the spatial weight matrix. There exist several common spatial-temporal weight functions, such as the Distance threshold function, the Inverse distance function, and the Gauss function. Among them, the Gaussian function is typically favored for GWR and GTWR modelling. Consequently, it is employed in this study and can be represented by Eq. (5).

$$w_{ij} = e^{\left[ -(d_{ij}/b)^2 \right]} \tag{5}$$

Where, b represents the bandwidth which is a non-negative constant for explaining the functional relationship between $w_{ij}$ and $d_{ij}$, while $d_{ij}$ refers to the Euclidean distance between samples $i$ and $j$. The value of Guass function decreases with the increase of $d_{ij}$. When $d_{ij} = 0$, $w_{ij} = 1$ and reaches the maximum. When $d_{ij}$ is infinite, $w_{ij}$ is infinitely close to 0, indicating

that the data point has almost no influence on the regression point. The function becomes smoother as the bandwidth b increases. The degree of weight reduction lessens as the distance grows. In this study, the Gaussian function method is utilized to determine the weight function.

## 2.3. Bandwidth determination

The bandwidth of weight functions has a more pronounced impact on the GWR and GTWR models than the spatial-temporal weight functions. A higher bandwidth might result in an augmented deviation and a diminished difference between various regions during parameter estimation. Conversely, a smaller bandwidth could attenuate the validity and amplify the variance of parameter estimation. Methods such as Cross-Validation (CV), Akaike Information Criterion (AIC), and Bayesian Information Criterion (BIC) are primarily employed to ascertain the optimal bandwidth.

The cross-validation method is employed to determine the bandwidth in this article. K-fold cross-validation is a statistical analysis technique and is typically utilized to assess the performance of a model. The original data set is partitioned into k equal subsets. Among them, k - 1 subsets serve as the training set, which is used for training the model, and the remaining single subset is the test set, which is employed for validating the trained model. The same experiment is carried out k times to guarantee that each subset functions as a validation set in turn. Generally, k is greater than 2. When the amount of data is extremely small, k can be set as 3. Ordinarily, k is selected as 5 or 10. In this paper, k is equal to 5. Identification index of k-fold cross validation can be demonstrated as Eq. (6):

$$CV = \sum_{I=1}^{n} \left[ y_i - \check{y}_{\neq i}(b) \right]^2 \tag{6}$$

$\check{y}_{\neq i}(b)$ is the fitting value at the regression point i, which indicates that, the parameters at the regression point can be estimated by the samples around the regression point. The bandwidth b corresponding to the minimum CV value is precisely the optimal bandwidth. This research opts for the five-fold cross-validation method to ascertain the bandwidth.

## 3. Study area and data collection

### 3.1. Study area and division

Xianyang is situated in northwest China and is adjacent to the provincial capital Xi'an in the east within Shaanxi Province. It covers a total area of 10,196 square kilometers and has a permanent population of 3,959,842 with an annual GDP of 220.481 billion yuan. A built-up area of 60 square kilometers has been designated as the research area, which encompasses 630,000 people and 190 communities. The boundaries of the communities, serving as the research unit, are illustrated in Fig 1.

### 3.2. Data sources and processing

This study primarily investigated the relationship among cycling frequency, community socio-economic characteristics, and the built environment. The multi-source data encompassed Xianyang personal cycling survey data, road network characteristics, land use, as well as other social and economic characteristics, all of which were provided by the Xianyang Planning and Design Institute. The personal cycling survey was conducted in Xianyang from June 10th to 20th, 2022, yielding 22,616 pieces of cycling data.

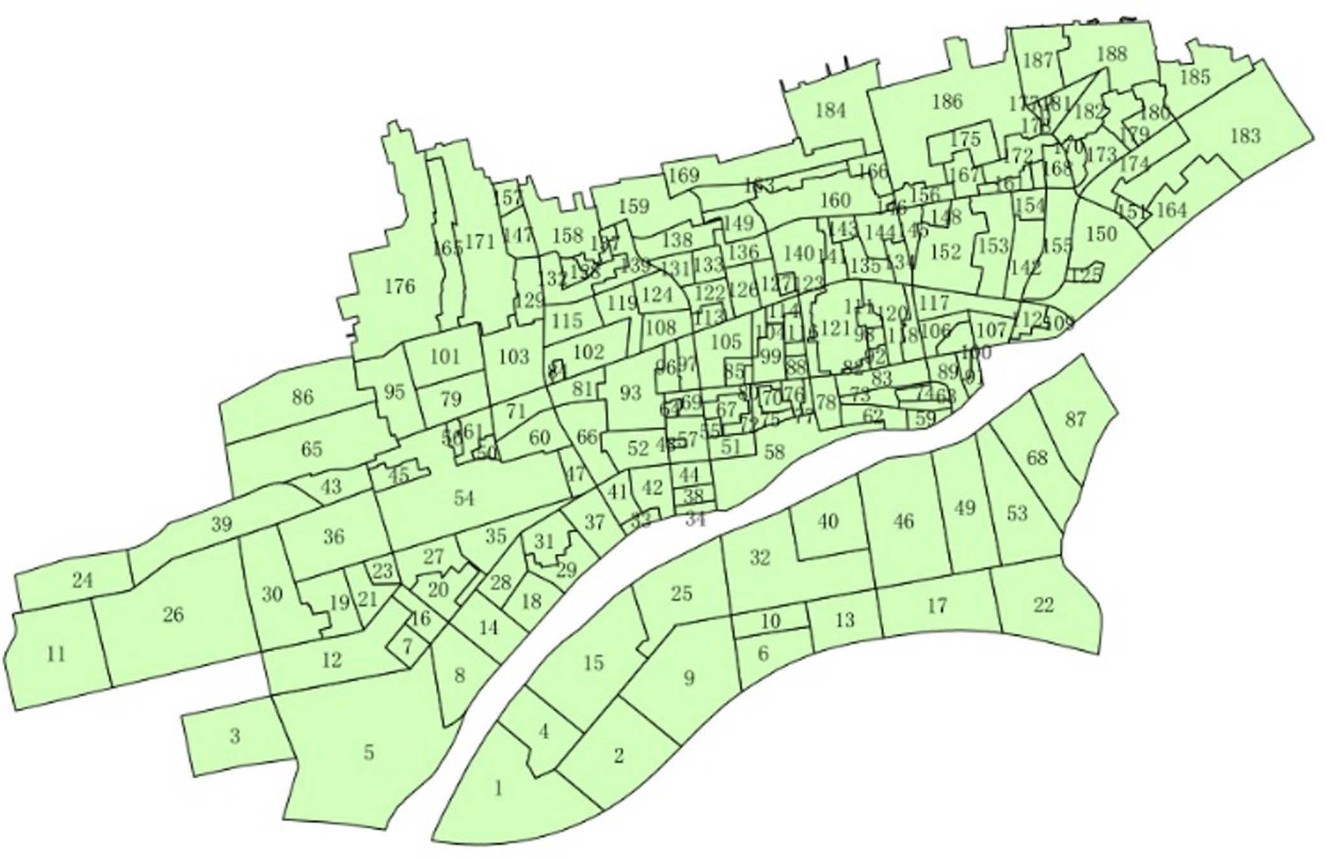

**Fig 1. The boundaries of the communities.**

The survey contents comprised gender, age, income level, car and bicycle ownership, riding start time, riding start position (latitude and longitude), riding arrival time, and riding arrival position (latitude and longitude). It is worth noting that, on the first page of the questionnaire, we introduced our research purpose and obtained their written informed consent to participate in this academic research. Non-adult participants were permitted to answer this questionnaire only after obtaining the written informed consent of their parent or guardian. Based on the survey results, the corresponding travel situations of each traffic zone are depicted in Fig 2 and Fig 3.

The average number of trips in each community is 80, and the maximum number of trips reaches 695. There are 9 traffic districts with more than 200 trips, accounting for approximately 20% of the total trips. The average riding distance is 2.28 kilometers, and the maximum riding distance is 5.58 kilometers. Around 82% of bicycle travelers have a riding distance of less than 3 kilometers. As can be observed from Fig 2, in terms of both travel frequency and travel distance, the bicycle travel demand in the urban fringe area is higher than that in the city center, with the travel frequency and travel distance being greater than those in the central area.

Fig 3 illustrates the distribution of departure and arrival times of bicycles. The peak of bicycle riding coincides with the peak hours of commuting, suggesting that bicycle riding is mainly for commuting and school trips. The peak hours of bicycle riding are 7 a.m. and 6 p.m.

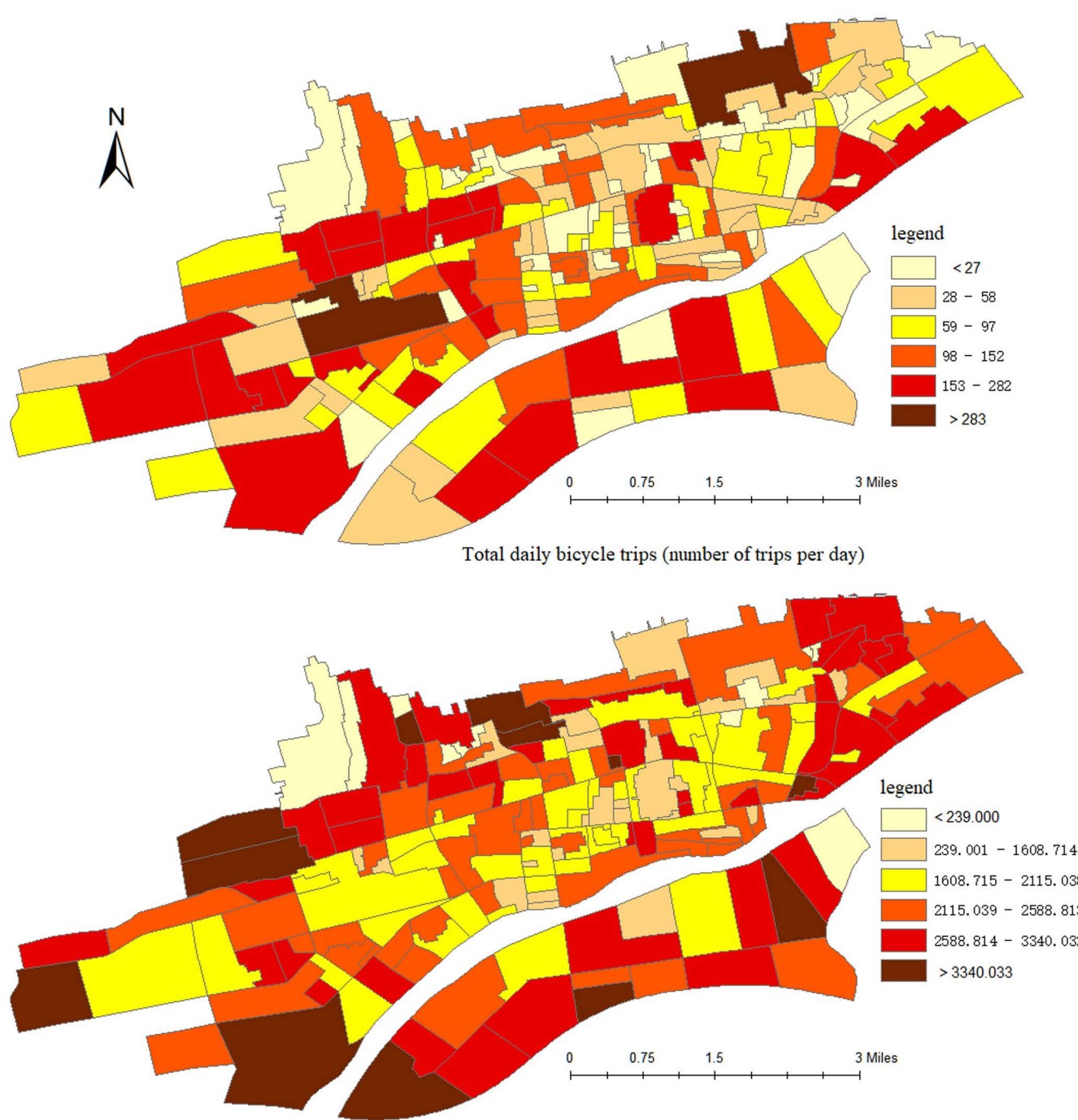

**Fig 2. The travel frequency and travel distance.**

Additionally, there is a minor travel peak at 12 noon and 2 p.m., which is associated with the scale of the city and commuting habits, and is in line with the time requirements for commuting to and from work and school.

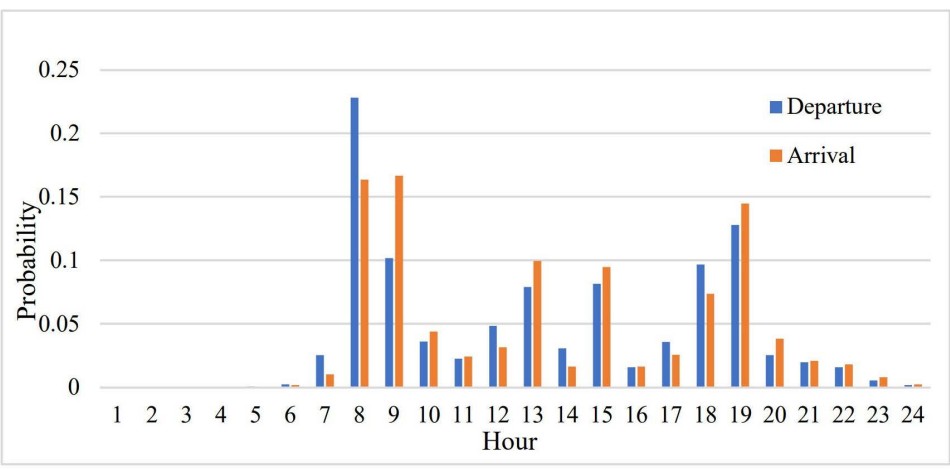

**Fig 3. Temporal distribution of cycling frequency.**

## 3.3. Variable selection and description

**3.3.1. Selection of independent variables.** According to current theories and related research, the factors that influence cycling can be categorized into three groups: socio-economic variables, built environment variables, and road traffic infrastructure variables.

Socio-economic variables encompass population density, gender, age, income, transportation costs, occupation, car and bicycle ownership, housing prices, etc. For detailed information, refer to Table 2.

Road factors comprise road network density, intersection density, bus station density, and riding safety factors. The riding safety facilities are evaluated by two indices, namely the proportion of the greenbelt between motorized lanes and non-motorized lanes and the proportion of non-motorized lane parking.

Environmental factors include the quality of seven types of POI (Point of Interest) facilities, the diversity of POI, the distance from the Central Business District (CBD), and whether the community is located in the CBD.

Owing to the diverse needs of residents, different types of POI facilities possess varying degrees of attraction for them. Consequently, this paper incorporates the preference coefficient of facilities to assess the quality of POI facilities from two perspectives: residents' preferences and the quantity of facilities. By referring to the approach of selecting facility categories in the research on the urban life convenience index by relevant scholars [34], the POI facilities are categorized into seven types: convenience services, catering facilities, education facilities, financial facilities, daily shopping, leisure and entertainment, and medical facilities. Through on-site interviews, the frequency and demand preferences of residents for commonly used facilities are examined. The Delphi method is employed to ascertain the weights of each category of facilities and their sub-categories via expert scoring and AHP (Table 1). Based on the aforementioned index weights, the calculation method for the residents' preference evaluation index of facilities is devised as Eq. (7):

$$C_i = \sum_j w_{ij} \cdot n_{ij} \tag{7}$$

Where, $C_i$ represents the quality of Class I POI facilities in level 1 factors, $w_{ij}$ represents the weight corresponding to the j-th type of secondary factor facilities of the i-th type of primary

**Table 1. Evaluation of Residents 'Preference for Facilities.**

| Variables | level 2 factors | $w_{ij}$ | Variables | level 2 factors | $w_{ij}$ |
|---|---|---|---|---|---|
| service | Domestic service | 0.102 | bank | ATM | 0.072 |
| | Beauty salons | 0.064 | | State-owned banks | 0.072 |
| | maintenance | 0.102 | | Other banks | 0.072 |
| restaurant | Fast-food restaurant | 0.05 | shopping | Department store | 0.027 |
| | western restaurant | 0.05 | | Convenience stores | 0.053 |
| | casual restaurants | 0.05 | | Chain supermarket | 0.053 |
| | Chinese Restaurant | 0.05 | | Farner's market | 0.018 |
| education | kindergarten | 0.031 | leisure | Culture and arts | 0.024 |
| | | | | Theatre, gymnasium | 0.064 |
| | Primary school | 0.074 | hospital | pharmacy | 0.028 |
| | | | | Community hospital | 0.022 |
| | Secondary school | 0.012 | | General, specialized hospital | 0.066 |

factor and $n_{ij}$ is the number of the j-th type of secondary factor facilities of the i-th type of primary factor.

The diversity of POI facilities is gauged by the entropy index, which is prevalently utilized in economics, and the formula is presented as Eq. (8):

$$L = \frac{-\sum_{j=1}^{k} p_j \cdot \ln(p_j)}{\ln k} \tag{8}$$

Where, $L$ represents the mixing degree of POI in the traffic zone, $P_i$ represents the ratio of the amount of class $j$ POI to the amount of all types of POI in the traffic zone, and $k$ represents the total amount of types of POI facilities in the traffic zone. $\ln(x)$ is logarithmic function based on the constante.

**3.3.2. Selection of dependent variables.** To analyze the spatial and temporal heterogeneity of cycling behavior, the travel frequency per unit area of bicycles (hereinafter abbreviated as travel frequency) is chosen as the dependent variable, which serves as a metric for cycling intensity. All independent variables denote the relevant indicators of the traffic area from which a bicycle trip commences, that is, the starting point of the trip. To determine the suitability of the model, a statistical distribution test is initially carried out on the dependent variable, resulting in a normal Q-Q plot as depicted in Fig 4. From the slope of the distribution, it can be seen that the travel frequency shows an approximately normal distribution, fulfilling the normality requirement for the dependent variable in a linear regression model.

The variables and indicators are presented in Table 2. The traffic zone is adopted as the research unit, and all the variables signify the specific index of the traffic zone.

**3.3.3. Multicollinearity diagnosis.** Regression models utilize multiple independent variables to account for a dependent variable. The correlation between each independent variable can have an impact on the explanatory power of the model, leading to the estimation results of the regression model deviating from reality or having low accuracy. Hence, it is essential to initially test for multicollinearity among the independent variables. In this part, the Variance Inflation Factor (VIF) is employed to evaluate the collinearity among the independent variables, as presented in Table 3.

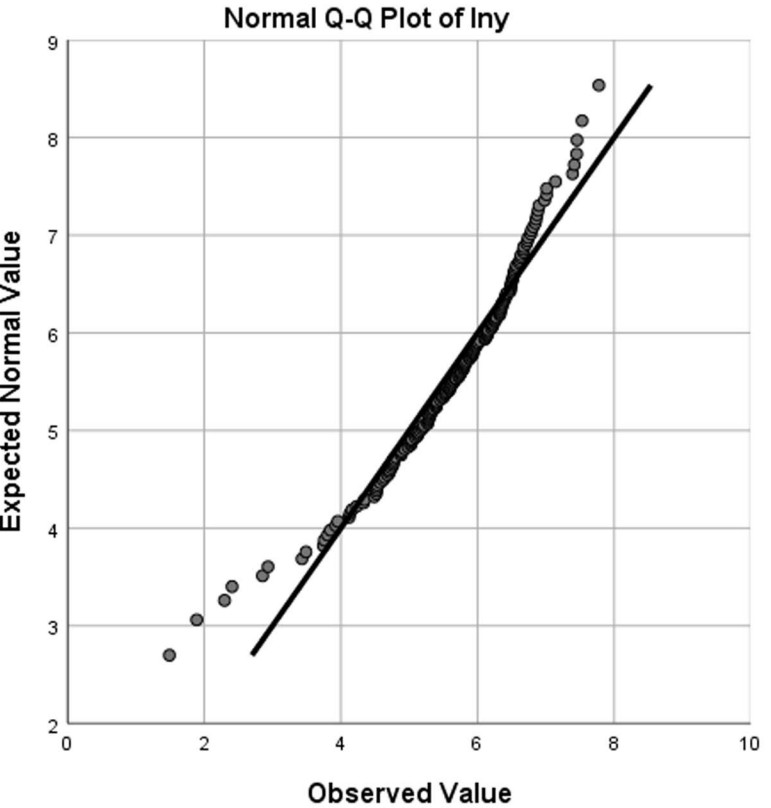

**Fig 4. The normal Q-Q plot.**

As can be observed from Table 3, the Variance Inflation Factor (VIF) values of the three categorical variables—gender, age, and occupation—exceed 5, signifying significant multi-collinearity. Thus, the aforementioned categorical variables were grouped and combined. Precisely, the gender variable is denoted by the ratio of males to females. The gender ratio is computed as the number of males divided by the number of females; a higher gender ratio implies a greater proportion of males. Age is represented by its average value, which is derived by multiplying the median of each age interval by its corresponding proportion and subsequently summing up these products. For the occupational distribution, six distinct weights are allocated to the six subtypes: civil servants = 1, corporate employees = 2, retirees = 3, unemployed = 4, students = 5, and others = 6. The weighted occupational value for each category is then calculated by multiplying the weight of the occupation by its corresponding population. A higher weighted occupational value indicates that the residents of the community are more prone to having flexible job types, such as being unemployed or self-employed; a lower value suggests that residents are predominantly employed in corporate or governmental positions.

The results of the multicollinearity test for the combined variables are shown in Table 4.

The results of the collinearity test demonstrate that all variables fulfill the requirements following the merger. Nevertheless, the relevant findings indicated that the two variables of gender and occupation were not statistically significant. Consequently, these two variables were excluded from the subsequent analysis. There are a total of 23 variables involved in this paper. The descriptive statistics of the various independent variables are presented in Table 5.

**Table 2. Description of variables and indicators.**

| Categories Variables | | Indicators Description |
|---|---|---|
| Dependent variable Y (The amount of bicycle trips per unit area (times/ha) of the traffic zone) | | Total daily bicycle trips in the traffic zone/ Area of the traffic zone |
| Socio-economic factors | Population | The population density of the traffic zone (person/ ha) |
| | Gender | Probability of male(female) of the traffic zone |
| | Age | Probability of people within the according group (<18,18–35,36–45,46–60,>60) of the traffic zone |
| | Job | Probability of people belong to each occupation (Students/Civil servants/Employees/ Unemployed persons/ retirees) of the traffic zone |
| | Employment | Probability of employed population of the traffic zone |
| | Car | Car ownership per household of the traffic zone |
| | Bicycle | Bicycle ownership per household of the traffic zone |
| | Income(<4000)/Income[4000–10000]/Income(>10000) | Monthly household income probability of the traffic zone |
| | Cost(<300)/cost[300,1000]/cost(>1000) | Monthly household traffic expense probability of the traffic zone |
| | Household area | Probability of per capita housing area of the traffic zone |
| | Price | Average housing price of the traffic zone(103 yuan) |
| Built-up environment factors | Quality of POIs(service, restaurant, education, hospital, shopping, bank, leisure) | Quality of each POI for the traffic zone, formula (1) |
| | Mixture | Mixture of land use for each traffic zone, formula (2) |
| | Distance | Distance from the center point of the traffic zone to the urban CBD (103 m) |
| | CBD | Whether the traffic zone is CBD, Yes-1. No-0 |
| Road factors | Road | Density of road network (km/ ha) of the traffic zone |
| | Intersection | Intersection density (number/ ha) of the traffic zone |
| | Bus | Bus station density (number/ ha) of the traffic zone |
| | Park | Non-motorized lanes' length which is invaded by roadside parking in the traffic zone/ Area of the traffic zone |
| | Isolation | Length of road which is isolated by green belt between motorized lane and non-motorized lane in the traffic zone/ Area of the traffic zone |

**Table 3. Results of the multicollinearity test for the independent variables.**

| Variable | Tolerance | VIF | Variable | Tolerance | VIF |
|---|---|---|---|---|---|
| The proportion of females | 0.07 | 15.05 | Convenience service facilities | 0.36 | 2.78 |
| Age [18–35] | 0.05 | 21.04 | Catering facilities | 0.43 | 2.31 |
| Age [36–45] | 0.10 | 10.23 | Educational facilities | 0.60 | 1.67 |
| Age [46–60] | 0.07 | 14.82 | Financial facilities | 0.36 | 2.74 |
| Age (>60) | 0.16 | 6.18 | Daily shopping facilities | 0.24 | 4.09 |
| Work | 0.06 | 5.89 | Leisure and entertainment facilities | 0.30 | 3.36 |
| Retire | 0.65 | 1.03 | Medical facilities | 0.58 | 1.73 |
| Unemployed | 0.44 | 2.27 | POI Diversity | 0.41 | 2.45 |
| Student | 0.06 | 18.12 | Distance from the City Center | 0.52 | 1.92 |
| Others | 0.62 | 1.61 | Road Network Density | 0.26 | 3.91 |
| Employment rate | 0.33 | 3.07 | Parking Proportion on Non-motor Vehicle Lanes | 0.64 | 1.56 |
| Average number of cars per household | 0.35 | 2.86 | Separation Proportion between Motor and Non-motor Vehicles | 0.49 | 2.06 |
| Average number of bicycles per household | 0.57 | 1.75 | Intersection Density | 0.29 | 3.48 |
| Average income | 0.56 | 1.78 | Bus Stop Density | 0.67 | 1.5 |

Note: The proportion of males, age (<18), with civil servants serving as the reference group.

**Table 4. Collinearity diagnostics.**

| Variables | Correlation | VIF | Variables | Correlation | VIF |
|---|---|---|---|---|---|
| Population | 0.551** | 1.19 | Bank | 0.359** | 1.24 |
| Age | 0.185* | 4.00 | Shopping | 0.377** | 3.47 |
| Gender | -0.042 | 1.19 | Leisure | 0.26** | 2.13 |
| job | -0.013 | 1.66 | Hospital | 0.41** | 1.24 |
| Employment | 0.217** | 3.95 | Mixture | 0.228** | 1.71 |
| Income | 0.222** | 4.19 | Distance | -0.314** | 1.65 |
| Car | 0.161* | 4.07 | CBD | 0.269** | 1.74 |
| Bicycle | 0.238** | 1.88 | Road | 0.294** | 2.54 |
| Cost | 0.175* | 4.21 | Park | 0.192** | 1.15 |
| Price | 0.184* | 1.06 | Isolation | 0.196** | 1.65 |
| Service | 0.417** | 2.11 | Intersection | 0. 166* | 2.38 |
| Restaurant | 0.385** | 2.23 | Bus | 0. 179* | 1.13 |
| Education | 0.143* | 1.21 | | | |

Note: ** = significant at the 0.01 level. * = significant at the 0.05 level.

**Table 5. Descriptive statistics of explanatory variables.**

| Variable | Average Value | Standard Deviation | Maximum Value | Minimum Value |
|---|---|---|---|---|
| Population | 213.25 | 374.86 | 3366.59 | 0 |
| Age | 35.65 | 10.97 | 53 | 0 |
| Employment | 0.76 | 0.27 | 1 | 0 |
| Income | 7.58 | 2.6 | 13.17 | 2 |
| Car | 0.52 | 0.3 | 1.22 | 0 |
| Bicycle | 0.74 | 0.43 | 2.9 | 0 |
| Cost | 3.23 | 1.73 | 9 | 0 |
| price | 7.745 | 6.328 | 92.000 | 4.096 |
| Service | 17.05 | 25.82 | 189.51 | 0 |
| Restaurant | 18.87 | 32.4 | 232.97 | 0 |
| Education | 1.93 | 3.31 | 23.86 | 0 |
| Bank | 12.47 | 23.88 | 190.64 | 0 |
| Shopping | 12.42 | 21.65 | 228.31 | 0 |
| Leisure | 3.89 | 7.49 | 52.62 | 0 |
| Hospital | 5.64 | 8.02 | 52.54 | 0 |
| Mixture | 0.64 | 0.26 | 0.97 | 0 |
| Distance | 2.716 | 1.622 | 8.502 | 0 |
| CBD | 0.234 | 0.423 | 1.000 | 0 |
| Road | 36.03 | 25.57 | 115.78 | 0 |
| Park | 6.49 | 12.27 | 93.04 | 0 |
| Isolation | 0.25 | 0.63 | 6.29 | 0 |
| Intersection | 0.15 | 0.21 | 1.16 | 0 |
| Bus | 0.04 | 0.08 | 0.59 | 0 |

# 4. Results

## 4.1. Exploratory spatiotemporal analysis

### 4.1.1. Spatial exploratory analysis.

(1) Global Spatial Autocorrelation Test for Travel Frequency

Spatial autocorrelation analysis is mainly utilized to investigate the spatial correlations between the sample data of a specific location and those of other adjacent samples. A positive spatial autocorrelation implies that the changing trend of a certain sample is in line with that of the adjacent samples. A significant spatial correlation serves as an important prerequisite for the analysis of spatial-temporal heterogeneity when the GTWR model is employed. If the results of the spatial correlation analysis suggest that the independent variables are not significant, it means that the geographical differences would have a minimal impact on the dependent variables, and other regression models should be considered for further research. Generally, Moran's I index is used for spatial autocorrelation analysis, as presented in Table 6.

The results of the autocorrelation analysis of the dependent variable are presented in Fig 5. The findings revealed that among the socio-economic attributes, the spatial autocorrelation of age and employment ratio was not significant, suggesting that the spatial distribution of these independent variables was random. Consequently, these two types of variables were excluded from subsequent modeling. The significance levels of the remaining variables met the requirement of $p < 0.005$, indicating that these variables exhibited spatial correlation. With the

**Table 6. Global spatial autocorrelation analysis.**

| Alternative independent variable | Moran's I | z-value | p-value |
|---|---|---|---|
| Travel density | 0.21 | 4.92 | 0 |
| Population | 0.09 | 2.57 | 0.02 |
| Age | 0.06 | 1.43 | 0.15 |
| Employment | 0.02 | 0.62 | 0.53 |
| Income | 0.11 | 2.49 | 0.01 |
| Car | 0.18 | 4.13 | 0 |
| Bicycle | 0.15 | 2.77 | 0.01 |
| Cost | 0.11 | 2.6 | 0.01 |
| Price | 0.04 | 4.06 | 0 |
| Service | 0.19 | 4.51 | 0 |
| Restaurant | 0.29 | 6.93 | 0 |
| Education | 0.05 | 2.17 | 0.04 |
| Bank | 0.11 | 2.7 | 0.01 |
| Shopping | 0.34 | 9.2 | 0 |
| Leisure | 0.34 | 7.99 | 0 |
| Hospital | 0.06 | 2.18 | 0.04 |
| Mixture | 0.18 | 4.06 | 0 |
| Distance | 0.93 | 26.29 | 0 |
| CBD | 0.64 | 18.07 | 0 |
| Road | 0.09 | 2 | 0.05 |
| Park | 0.17 | 4.06 | 0 |
| Isolation | 0.13 | 3.4 | 0 |
| Intersection | 0.08 | 2.18 | 0.04 |
| Bus | -0.09 | -2.57 | 0.02 |

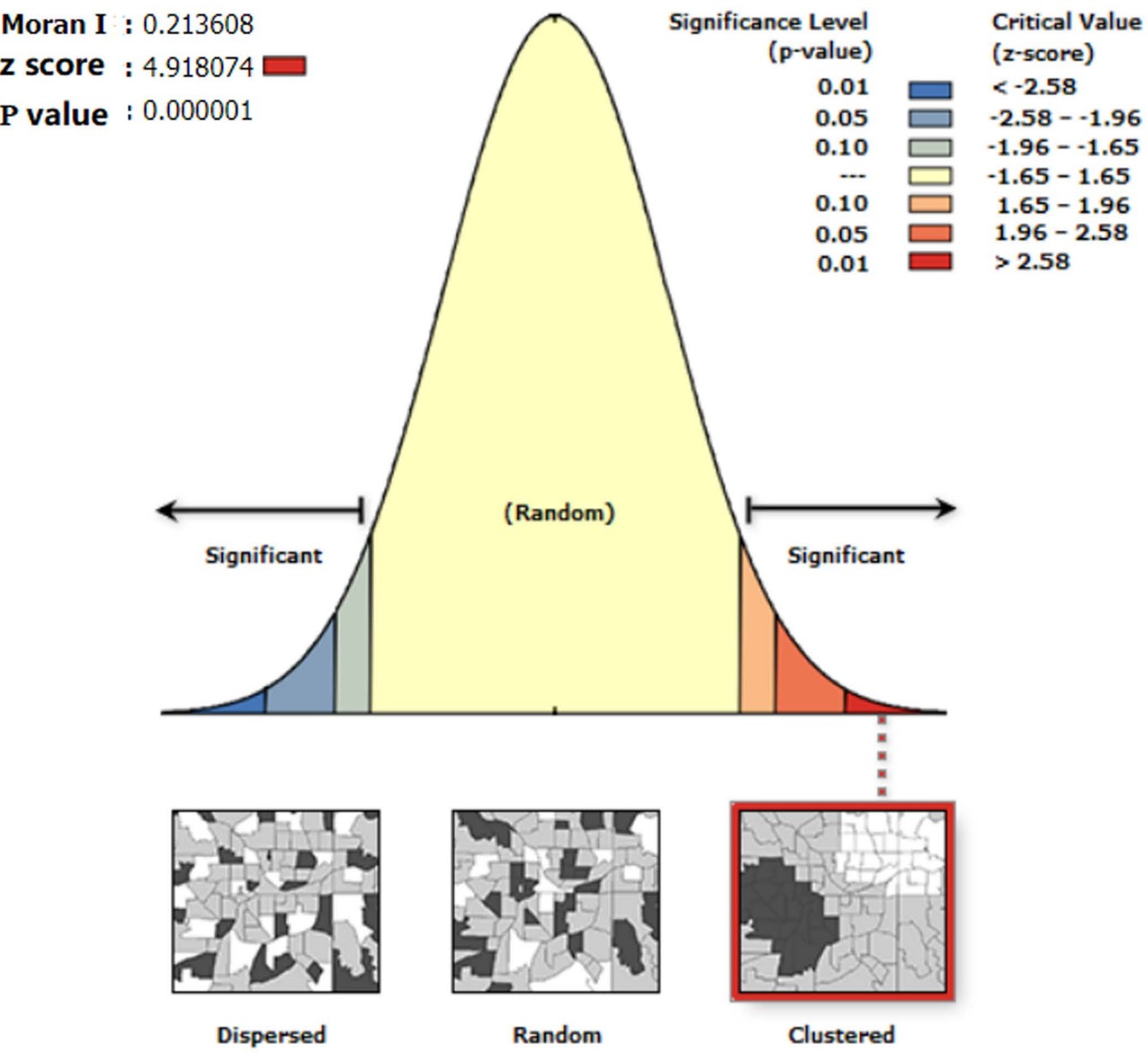

**Fig 5. The distribution of Moran's I statistic.**

exception of the variable of bus station density, both Moran's I and the aggregation characteristic value Z were positive, signifying that most variables were strongly spatially agglomerated.

(2) Local Spatial Autocorrelation Test for Travel Frequency

The Moran Scatter Plot is utilized to depict the relationship between the bicycle riding frequency (expressed as a deviation from the mean) and the mean of the spatial lag of bicycle riding frequency (that is, the weighted average of bicycle riding frequencies corresponding to other traffic zones adjacent to a particular traffic zone). Fig 6 and Fig 7 illustrate the Local Indicators of Spatial Association (LISA) maps for the local spatial autocorrelation of bicycle riding frequency.

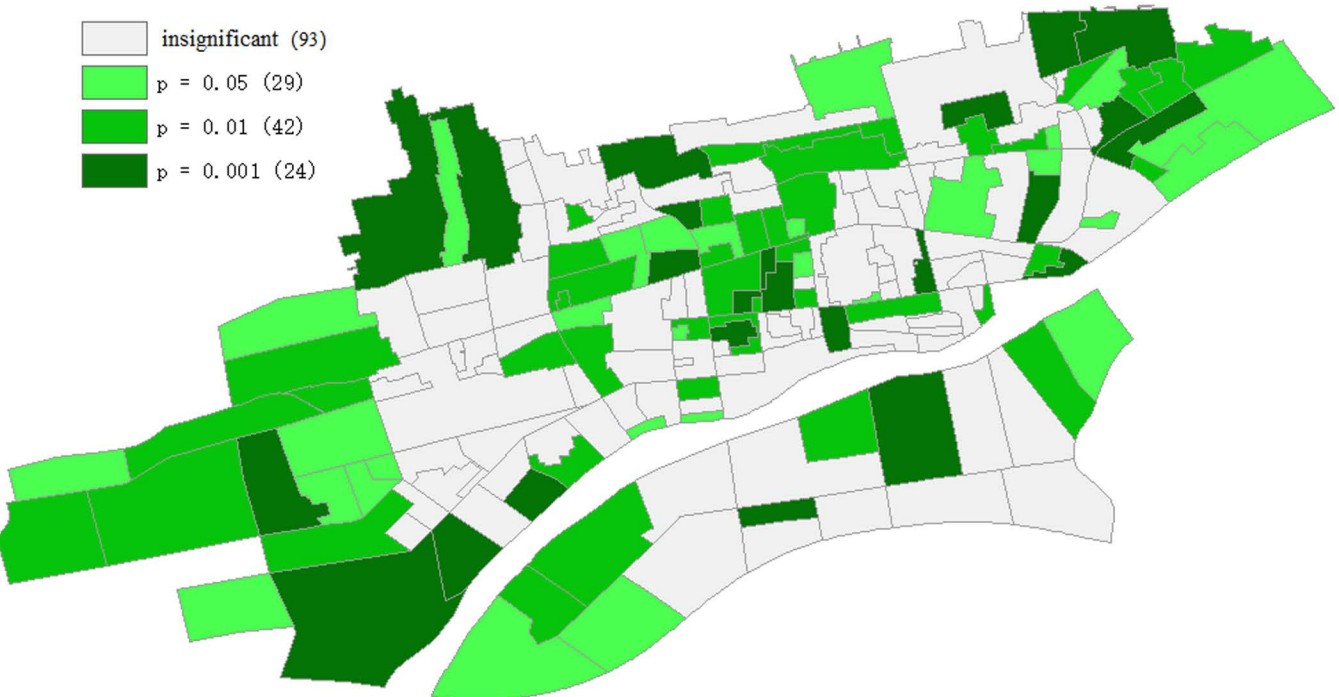

Local Spatial LISA Significance Map for Travel Frequency

**Fig 6. Local spatial LISA significance map for travel frequency.**

Fig 7 shows the clustering relationships of cycling frequency among different traffic zones at the 5% confidence level. As depicted in the Fig 7, a substantial number of zones display a random distribution of cycling frequency. High-high clustering is mainly witnessed in the central area of Xianyang City, whereas low-low clustering is predominantly located in the eastern and western parts of the city. The central area enjoys convenient transportation and a high degree of land use mixing, which leads to relatively short daily travel distances for residents. Hence, the likelihood of cycling as a means of transportation is greater, and adjacent traffic zones possess similar travel characteristics, giving rise to a high-high clustering distribution. The western and eastern parts are newly developed areas, so the cycling frequency is lower, forming large low-low clustering regions.

**4.1.2. Time exploratory analysis.** The temporal variation pattern pertains to the alteration in the total bicycle riding frequency over a 24-hour period within the research area, as illustrated in Fig 8. It is clearly observable that 7:00 and 18:00 are the two peak periods for bicycle riding frequency, which typically correspond to the commuting and school travel times of residents. From 22:00–6:00, the quantity of bicycle rides is extremely low, which is also in line with the traits of bicycle travel.

## 4.2. Model construction and selection

The GWR model and the GTWR model are constructed respectively to compare the merits and demerits in analyzing the spatial heterogeneity of cycling. The goodness-of-fit of the two models is presented in Table 7.

The comparison of the results of the two models demonstrated that the GTWR model exhibits better performance in terms of goodness-of-fit. This is in line with the research

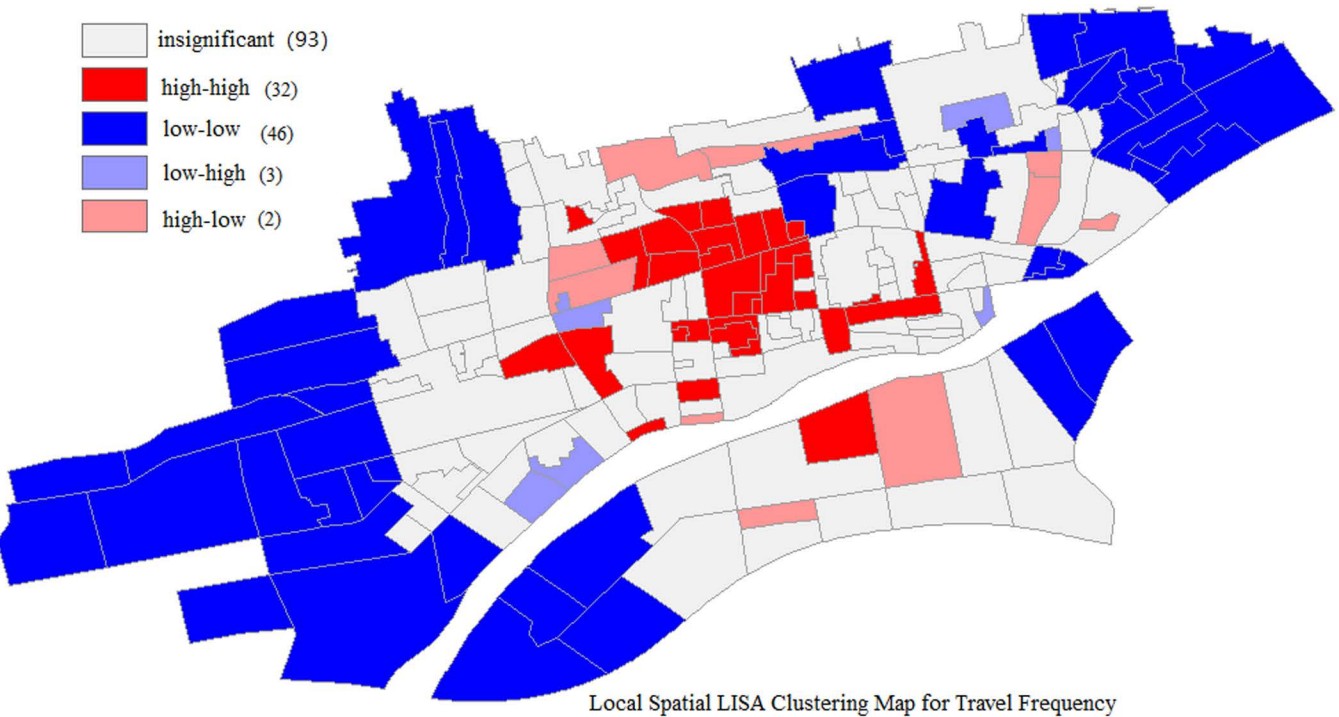

Local Spatial LISA Clustering Map for Travel Frequency

**Fig 7. Local spatial LISA clustering map for travel frequency.**

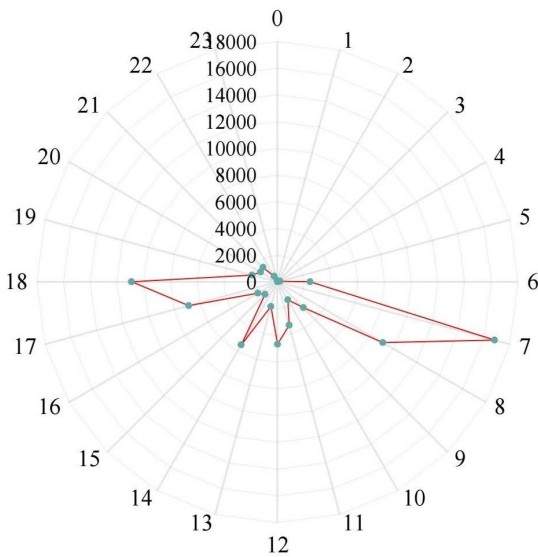

Daily Variation Pattern of Total Bicycle Riding Frequency

**Fig 8. Daily variation pattern of total cycling frequency.**

findings of Fotheringham and Yao, the developers of the GTWR model [35]. Subsequently, the GTWR model is employed to investigate the spatial and temporal distribution of cycling ridership. The model estimation results are presented in Table 8.

## 4.3. Spatial characteristic of variables based on GTWR

This paper places greater emphasis on the influence of the built environment on the spatial and temporal heterogeneity of cycling frequency. Subsequently, some representative variables are selected for spatial heterogeneity analysis, as depicted in Fig 9 and Fig 10.

The spatial heterogeneity of socio-economic factors can be elucidated as follows:

(1) Population density of the traffic zones

Overall, population density has a relatively weak promoting effect on the frequency of bicycle riding. However, in newly-built areas, it holds significant travel potential. Thus, an increase in population density in such areas will substantially boost bicycle riding.

**Table 7. The fitness of GWR model and GTWR model.**

| Variable | GWR | GTWR |
|---|---|---|
| R2 | 0.458 | 0.710 |
| adjusted R2 | 0.392 | 0.707 |
| AICc | 1530.825 | 1276.902 |
| residual sum of squares | 453.822 | 357.365 |

**Table 8. Estimation of GTWR models.**

| Variables | Minimum Value | Lower Quartile | Median | Upper Quartile | Maximum Value | Average Value |
|---|---|---|---|---|---|---|
| Population | -7169.08 | -0.08 | 5.61 | 48.60 | 16345.30 | 24.18 |
| Income | -1253.68 | -2.33 | 0.77 | 17.67 | 3451.57 | 9.63 |
| Car | -492.17 | -13.56 | 0.00 | 6.50 | 2713.10 | -0.13 |
| Bicycle | -3600.53 | -0.35 | 4.59 | 33.97 | 2804.44 | 24.77 |
| Cost | -3642.73 | -13.91 | -0.18 | 7.17 | 855.30 | -6.83 |
| Price | -51.699 | 0.110 | 4.442 | 18.041 | 84.333 | 13.064 |
| Service | -25453.10 | -3.84 | 2.49 | 34.80 | 8655.77 | -4.33 |
| Restaurant | -3338.09 | -4.85 | 0.57 | 28.71 | 7516.61 | 15.86 |
| Education | -2961.16 | -5.26 | 0.03 | 17.42 | 4134.77 | 13.52 |
| Bank | -761.10 | -10.31 | 0.00 | 17.50 | 9056.57 | 28.02 |
| Shopping | -16291.40 | -24.16 | 0.00 | 42.35 | 3155.22 | 5.80 |
| Leisure | -592.88 | -0.48 | 5.51 | 35.12 | 5829.63 | 33.17 |
| Hospital | -4790.14 | -10.29 | -0.01 | 10.57 | 1998.05 | 0.88 |
| Mixture | -217.71 | -12.14 | -0.33 | 1.50 | 359.79 | -7.24 |
| Distance | -63.900 | -26.615 | -12.325 | -1.727 | 1.148 | -17.115 |
| CBD | -17.190 | -3.127 | -1.194 | -0.209 | 3.571 | -2.232 |
| Road | -1335.75 | -1.56 | 1.18 | 23.94 | 2234.37 | 19.09 |
| Park | -887.76 | -5.40 | 0.00 | 13.22 | 3319.83 | 10.03 |
| Isolation | -2714.31 | -14.03 | 0.00 | 18.88 | 1096.60 | -3.10 |
| Interaction | -10253.80 | -21.90 | -0.08 | 3.95 | 2073.98 | -18.33 |
| Bus | -3422.01 | -9.40 | -0.04 | 5.68 | 5402.25 | -1.84 |
| Intercept | -465.88 | -3.42 | 0.00 | 5.25 | 413.40 | 1.51 |

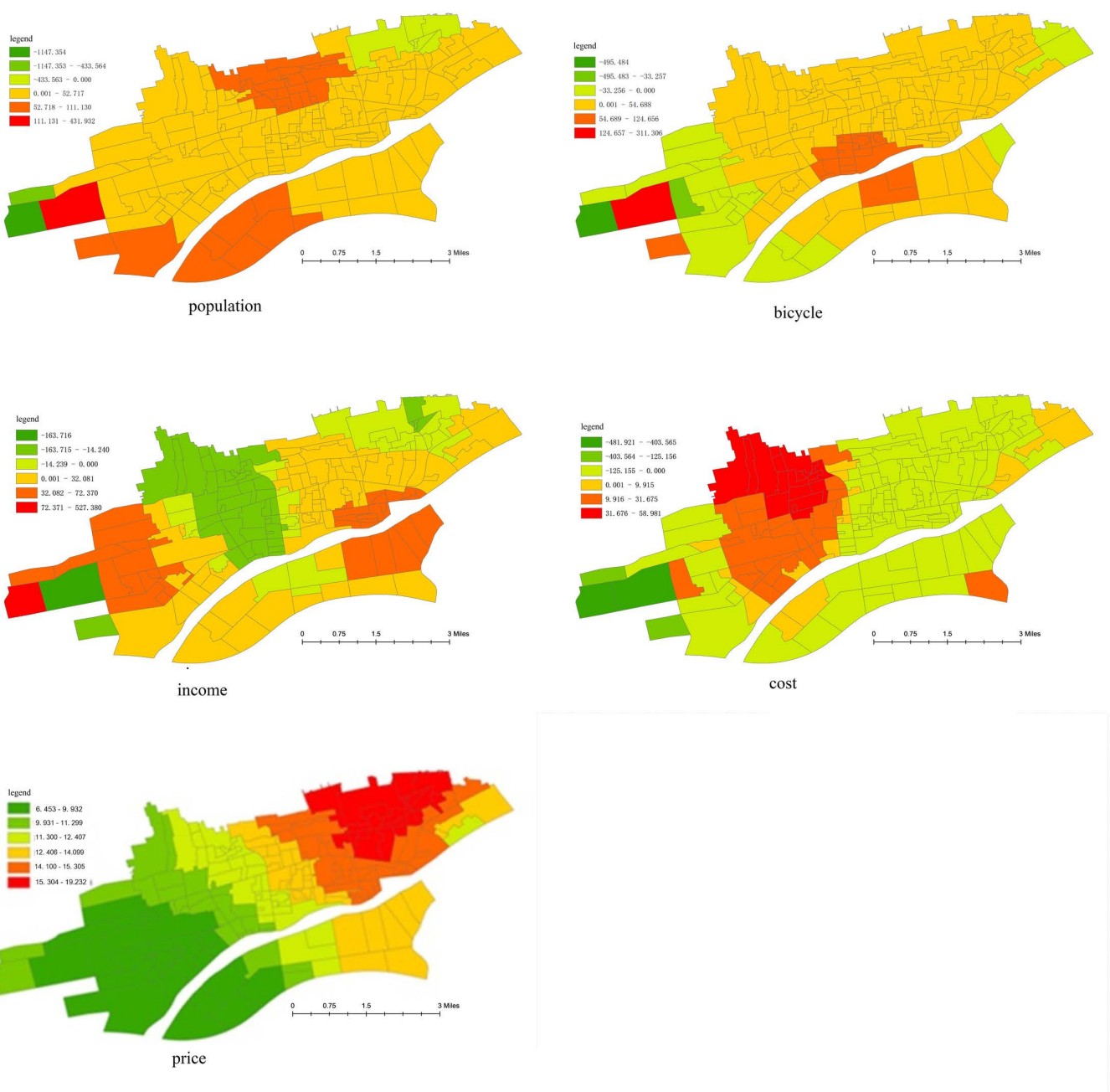

**Fig 9. The spatial heterogeneity of socio-economic factors.**

(2) Monthly income

In the western region, monthly income can enhance the frequency of bicycle riding. This is mainly because the western region is a newly-developed area where the current land use is predominantly rural. Urban development implies an increase in income, which will drive up the total number of trips, consequently, raise the frequency of bicycle trips. Nevertheless, in the urban central area and the northern cultural and educational area, the urban

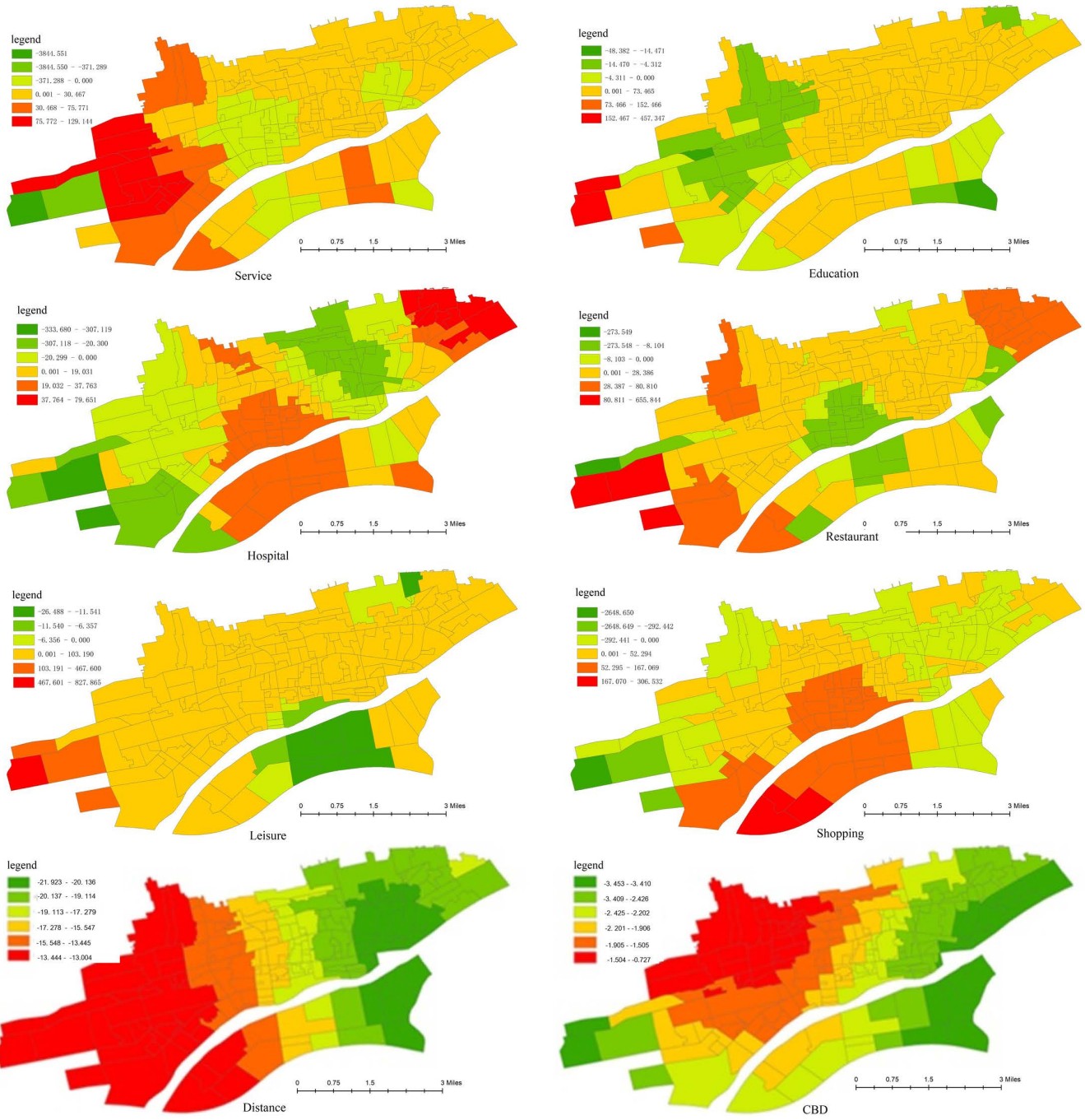

**Fig 10. The spatial heterogeneity of facilities variables.**

functions have been largely perfected, the employment rate is relatively high, and road traffic is approaching saturation. An increase in income will further elevate the employment rate, potentially leading to a simultaneous rise in motor vehicle trips. This, in turn, will exacerbate road traffic congestion and suppress the demand for bicycle trips.

(3) Average bicycle ownership per family

Evidently, an increase in bicycle ownership will significantly stimulate the cycling demand. The influence coefficients of each traffic zone are nearly identical, indicating that the impact of bicycle ownership on cycling demand is largely independent and is not influenced by other factors such as the built environment and road characteristics.

(4) Monthly average travel cost

The monthly average travel cost exerts an inhibitory effect on the demand for bicycle riding. An increase in the monthly average travel cost implies a higher proportion of motorized travel modes, such as private cars and taxis, thereby reducing the demand for bicycle travel.

(5) Housing price

The influence of average housing price on bicycle riding is closely associated with geographical location. The results reveal a negative impact of average housing price on bicycle riding in the western area, while it exhibits a strong promoting effect in the eastern area. This may be attributed to the higher diversity and mixing of POI in the eastern area compared to the western area. However, the average housing price in the western area is slightly higher than that in the eastern area, rendering the new population more mobile in the western area.

The spatial heterogeneity of facilities variables can be explained as follows:

(1) Convenience service facilities

The influence of the quality of convenience service facilities on cycling demand exhibits a gradually diminishing trend from the western to the eastern region. This may be attributed to the scarcity of convenience service facilities in the western area.

(2) Educational and medical facilities

The enhancement of the quality of educational facilities primarily promotes the travel of students who use bicycles. The impact of educational institutions in the central areas of the city is more pronounced than in other zones. Most schools are situated near residential areas, and the educational facilities in the central areas are relatively comprehensive. Further improving the quality of education can augment the proportion of students attending nearby schools, thereby spurring a significant increase in students' cycling.

The medical institutions in the suburbs of the city are mainly newly-established community clinics. Thus, the addition of large specialized hospitals will boost travel demand. The medical facilities in the central area are relatively well-developed, and all daily medical needs can be largely satisfied. Consequently, expanding the scale of medical institutions has a minimal effect on cycling. Overall, the scale of educational and medical institutions has a lesser influence on bicycle riding compared to other factors.

(3) Catering facilities

The impact of catering facilities on cycling demand is analogous to that of convenience service facilities. These two types of facilities are the principal daily destinations for residents. Improving the quality of such facilities can effectively stimulate the demand for bicycle travel.

(4) Leisure and shopping facilities

The influence of leisure facilities on cycling demand is congruent with that of shopping facilities, owing to the similar functions of the two types of facilities. Augmenting the quality of leisure and shopping facilities will substantially augment bicycle trips in the western and

southern parts of the Weihe River. This may be due to the insufficiency of facilities and a lower degree of land use mixing in these areas.

(5) Distance from CBD and whether it is CBD

The Xianyang CBD is located in the center of the study area, so the influence coefficient of the central area is the smallest. The zones closer to the urban fringe display a more conspicuous impact, signifying that cycling demand is closely correlated with the distance from the CBD. The western region has a promoting effect on cycling demand, and the impact is significant. The eastern region is an industrial area and exhibits an inhibitory effect. The farther away from the CBD, the smaller the proportion of bicycle trips.

The CBD and its surrounding areas have a notable promoting effect on bicycle travel, while the outer edge of the city has an inhibitory effect. This may be because the travel distance in the vicinity of the CBD is within a reasonable cycling range, whereas away from the edge of the urban CBD, the travel distance is long, and residents may opt for bus or private car travel.

(6) Road network density

The density of the road network has a positive impact on cycling frequency, implying that the improvement of road conditions can effectively attract cycling demand. However, in the area south of the Weihe River, the opposite tendency emerges. This may be due to the long cycling distance of residents in the south. The enhancement of road network density is accompanied by an improvement in bus service levels, and some cyclists will switch to using buses for commuting.

(7) Proportion of non-motor vehicle lanes occupied by parking

The proportion of non-motor vehicle lanes occupied by parking has an inhibitory effect on the frequency of cycling. The degree of inhibition progressively intensifies from the urban center to the periphery. This is mainly influenced by geographical location and the characteristics of bicycle travel. In the central area, parking encroachment has a limited impact on travel experience due to the short travel distance. In the suburbs, residents' travel distance is long, and the occupation of non-motor vehicle lanes will severely affect the comfort and safety of bicycle riding. Therefore, the more severe the occupation of non-motor vehicle lanes, the more likely residents are to abandon bicycle riding.

(8) The proportion of isolation belt between motorized and non-motorized lanes

The proportion of isolation belt between motorized and non-motorized lanes can enhance cycling frequency, especially in the urban fringe area. This is presumably because the installation of non-motorized isolation facilities can improve riding comfort and safety.

(9) Density of intersections

Intersection density has a significantly inhibitory effect on cycling frequency. One possible reason is that intersections disrupt the continuity and safety of bicycle riding, and a high density of intersections leads to an increase in riding time. Another reason could be that an increase in intersections implies more bus stops, causing some cyclists to transfer to public transportation. Additionally, the effect of intersection density on cycling frequency is less than that of parking encroachment and isolation belt. However, due to the low density of the road network in the southern region, an increase in intersection density can enhance travel accessibility and path selection diversity, thereby promoting bicycle riding in this area.

(10) Bus stop density

The density of bus stops has an inhibitory effect on the number of cycling demands, given the evident competition between buses and bicycles in short-distance travel. The improvement of

public transport service facilities will attract some potential bus travelers. Particularly in the marginal areas in the western part of the city, the enhancement of public transport facilities makes cyclists more prone to abandon cycling due to the long travel distance.

## 4.4. Temporal heterogeneity

The variation in the average value of the fitting coefficient of explanatory variables in the time dimension is depicted in Fig 11 and Fig 12.

Regarding social and economic attributes, the temporal influence of population density, income, and bicycle ownership on bicycle trips is comparable. These three attributes all possess a positive promoting effect. Around 8 AM, the promoting effect attains its maximum, and subsequently, the effect gradually fluctuates and weakens. The impact of these three factors on bicycle trips aligns with the morning peak hours, suggesting that the influence of social-economic attributes on bicycle trips is predominantly manifested in commuting trips. Travel cost exerts an inhibitory effect on cycling demand throughout the day, and its inhibitory effect exhibits three minor peaks, which occur around 8 AM, 12 PM, and 7 PM respectively. This influence reflects the three commuting peak hours. Housing price exhibits a promoting impact on bicycle riding throughout the day, with its most prominent effect during the 7–8 AM morning peak hour.

(1)  Convenience service facilities

Convenience service facilities exert an inhibitory influence on bicycle trips prior to 7 PM. The promoting effect reaches its zenith at 9 PM. This is because trips to service facilities such as domestic services, beauty salons, and other service facilities mostly occur between 7 PM and 10 PM.

(2)  Educational facilities

The promoting effect of educational facilities on cycling demand coincides with the commuting peak period. The maximum effect emerges at 8 AM and 6 PM, signifying that bicycles are one of the principal commuting tools in Xianyang City.

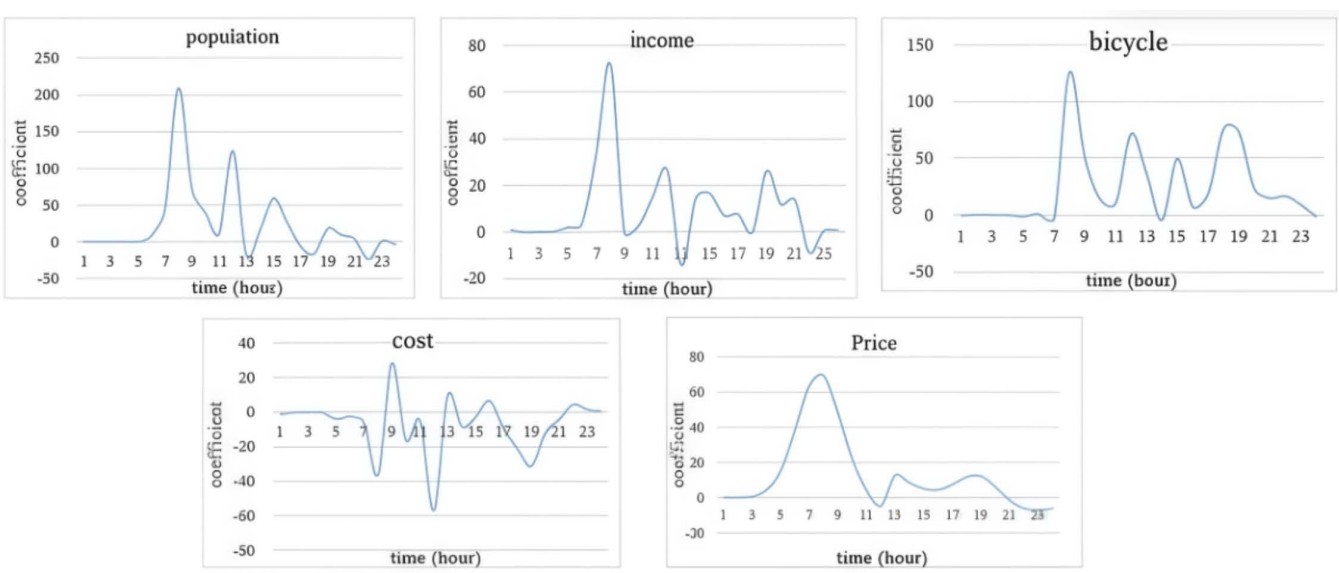

**Fig 11. The influence of socio-economic variables in the time dimension.**

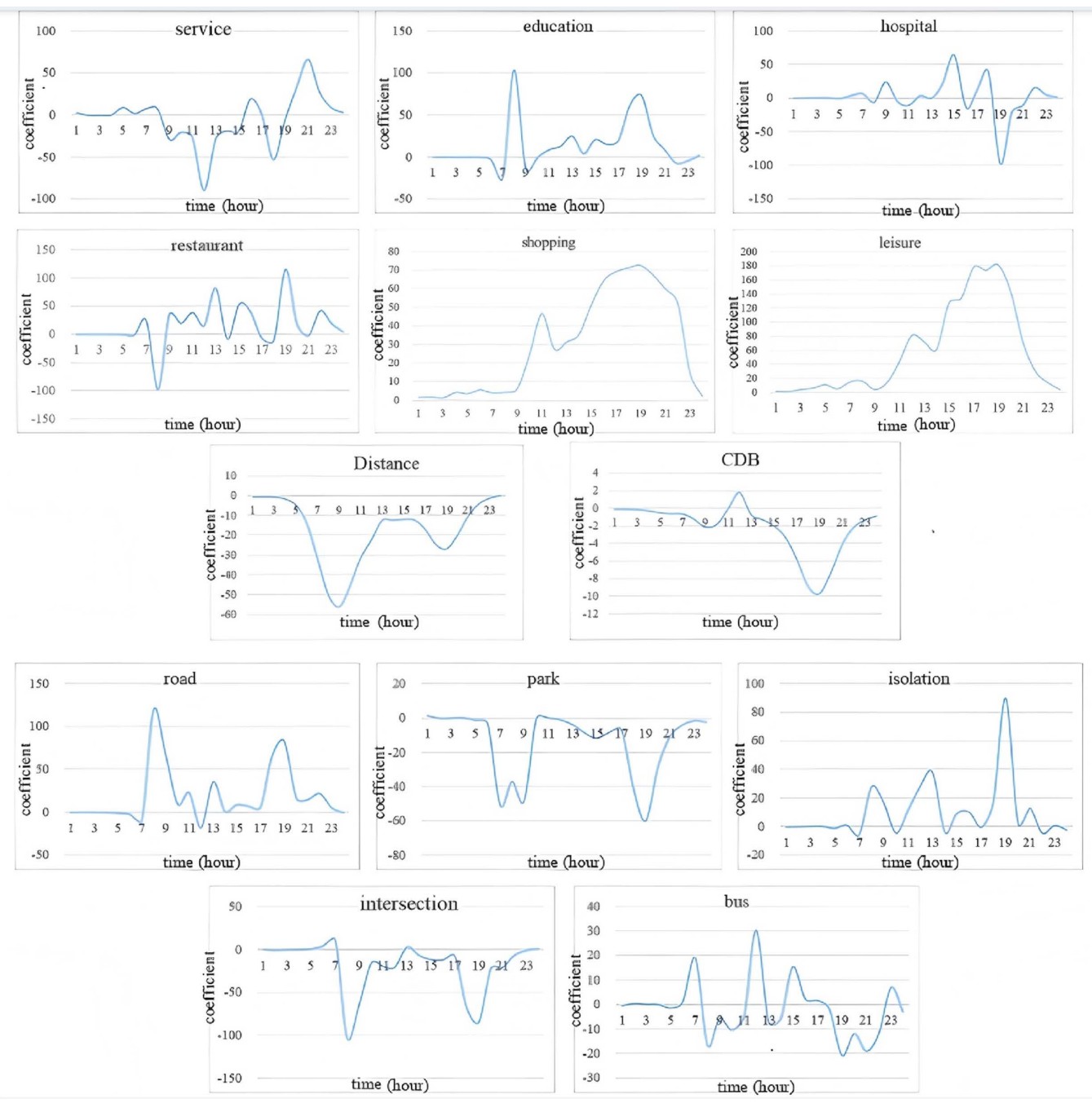

**Fig 12. The influence of facility variables in the time dimension.**

(3) Medical facilities

Medical facilities have a feeble promoting effect on cycling demand, indicating that bicycles are not the primary means for residents to seek medical treatment. Particularly between 7 PM and 9 PM, medical facilities have a significant inhibitory effect on cycling demand. This is because most night emergency patients are transported by vehicle.

(4) Catering facilities

The influence of catering facilities on bicycle trips transitions from inhibition to promotion. At 8 AM, the main objective of residents' travel is commuting, so there are very few trips to catering facilities, thereby demonstrating a restraining effect. After 11 AM, the positive promoting effect of catering facilities becomes evident as trips for dining purposes increase, especially at 1 PM and 7 PM. The results are in accordance with residents' daily living habits.

(5) Shopping and leisure facilities

The peak value of the influence coefficient of shopping and leisure facilities on bicycle travel lags behind that of commuting, which is related to residents' daily living habits. Daily trips for shopping and leisure predominantly occur between 6 PM and 10 PM, thus the peak correlation coefficients between 7 PM and 9 PM. Additionally, prior to 10 AM, shopping and leisure facilities scarcely affect the cycling demand. However, the promoting effect gradually intensifies after 10 AM. This is essentially the same as the operating hours of shopping and leisure facilities (which are generally 10:00–22:00).

(6) Distance from the CBD and whether it is CBD

The influence of distance and the CBD on bicycle riding has an inhibitory effect during the daytime, which is largely in line with the commuting peak hour. The inhibitory effect of distance on cycling is most conspicuous during the morning peak hours, indicating that the farther the distance, the smaller the proportion of bicycle trips. The influence of the CBD is most pronounced in the evening peak hours. This may be due to the fact that non-work trips in the CBD mostly occur in the evening peak hours after work and the heavy traffic during this period affects the cycling traffic space and safety.

(7) Road network density

Road network density has a promoting effect on cycling demand, and the promoting effect peaks around 8 AM and 7 PM. Before 7 AM, the augmentation of road network density does not promote bicycle trips, given the minimal demand for cycling at night and in the early morning. Moreover, the increase in road network density implies an improvement in bus service levels. Consequently, the enhancement of bus service levels will lead to the conversion of bicycle riding to bus travel during the period when bicycle travel demand is low.

(8) Proportion of non-motorized lanes occupied by parking

The extent of parking occupation of non-motorized lanes will impede the frequency of bicycle trips. The inhibitory effect is most prominent in the morning and evening peak hours, suggesting that when riding safety deteriorates, travelers will abandon cycling. Bicycle trips between 10 AM and 5 PM are less affected by parking occupation, and the influence coefficient fluctuates around 0. This is because cycling safety is less endangered during off-peak hours due to lower road traffic volumes.

(9) Proportion of road with isolation between motorized and non-motorized lane

The length of isolation between the motorized lane and the non-motorized lane can enhance the frequency of cycling. Before 6 AM, the isolation proportion has a negligible impact on the frequency of bicycle riding, given the minimal demand for cycling. However, after 6 PM, with the increase in the proportion of road isolation, the frequency of cycling gradually rises and reaches its peak at 7 PM.

(10) Intersection density

Intersection density has an inhibitory effect on cycling frequency, which is precisely the opposite of that of road network density. The effect is most conspicuous at 8 AM and 7 PM. In other periods, it has a minimal impact on cycling demand.

(11) Bus stop density

The density of bus stops has an inhibitory effect on bicycle riding during the morning and evening peak hours, which is primarily attributable to the competitive relationship between the bus and the bicycle. During the morning and evening peak hours, more individuals will tend to select bus travel due to the high frequency of bus departures and short waiting times. In contrast, during non-peak hours, residents are more inclined to choose bicycle travel, mainly because of the low frequency of bus departures and long waiting times.

## 5. Discussion and policy suggestions

The GTWR model reveals the heterogeneity in the spatial and temporal distribution of cycling demand and further corroborates the evidence of the correlation between the built environment and cycling frequency. The empirical results validate the aforementioned hypothesis that the quality of POI facilities and cycling safety facilities influence the spatial and temporal characteristics of cycling demand. Overall, socio-economic factors, built environment characteristics, and riding safety facilities all exhibit significant spatial and temporal heterogeneity in their impact on cycling frequency. Notably, an increase in the quantity and quality of facilities for daily life travel, such as convenience facilities, catering, and leisure shopping, can boost cycling demand. Road conditions, including a high density of road networks, a low proportion of non-motorized lane parking and a high proportion of isolation between motorized and non-motorized lanes, will also encourage bicycle trips.

In terms of spatial heterogeneity, the spatial effects of most variables are manifested by a gradual strengthening or weakening from the center to the periphery, reflecting the uneven distribution of facilities in urban center and edge areas. In particular, there is a deficiency in daily living facilities, such as convenience facilities, catering facilities and leisure shopping facilities in the western and southern areas with high population densities. However, the main impediment to bicycle travel in the central area is the lack of guaranteed cycling space and safety due to heavy traffic and high road saturation. Additionally, the proportion of non-motorized road parking and isolation facilities between motorized and non-motorized vehicles show a relatively consistent trend in the central and peripheral areas, that is, non-motorized vehicle parking will suppress cycling demand, while an increase in isolation facilities between motorized and non-motorized vehicles will promote cycling demand.

Regarding temporal heterogeneity, the temporal effect of each variable on cycling demand aligns with the time characteristics of commuting. The influence coefficient of most factors reaches its maximum during peak hours, indicating that the impact of built environment facilities on cycling frequency is mainly manifested in commuting. For shopping and leisure facilities, their promoting effect is consistent with flexible travel after work. This finding is in line with previous studies on the impact of bicycle travel and the built environment.

Based on the research results, the paper proposes the following policies and suggestions to promote the shift from motorized trips to bicycle trips:

Strengthen the construction and planning foresight of public service facilities and infrastructures in the western and southern areas. When the number of residents reaches a certain scale, ensure the quantity and scale of daily flexible travel facilities, including convenient

service facilities, catering, leisure shopping, and improve the density of road networks to meet the daily travel needs of surrounding residents within a cycling range.

Ensure that newly-built urban roads in the west and south of the city have sections of three or four lanes. Moreover, in the central area, when the saturation of motor vehicles on a particular road exceeds 0.65, timely install physical isolation facilities between motorized and non-motorized vehicles to ensure riding safety.

By continuously expanding high-quality education resources and standardizing enrollment methods, increase the proportion of students attending neighborhood schools and reduce the proportion of motorized trips to school by promoting the concepts of "High-Quality Resources" and "Fair Opportunities".

Construct new off-road public parking lots or implement off-peak sharing of parking resources to minimize the proportion of on-road parking; specifically, prohibit on-street parking on mixed road sections. Additionally, intensify the punishment for illegal parking in non-motorized lanes to ensure the availability of cycling space.

Implement off-peak commuting to alleviate traffic pressure during morning and evening peak hours in the central area, thereby increasing the space and safety of non-motorized riding and promoting cycling.

Non-motorized vehicle travel expenses mainly comprise private bicycle purchase costs and shared bicycle riding fees. Implement policies related to bicycle purchase discounts and riding coupons to encourage cycling.

Optimize the layout of bus stations, increase bicycle parking spaces at bus stations, transform the competitive relationship between "Bus and Bicycle" into a cooperative relationship of "Bus + Bicycle", and enable the bicycle to fulfill the function of the "Last Kilometer".

## 6. Conclusions

This paper constructs a GTWR model by utilizing multi-source data, such as Xianyang residents' cycling data and spatial geography data, to investigate the relationship among cycling frequency, socio-economic factors, public service facility quality and cycling safety facilities, and to account for the spatiotemporal heterogeneity of cycling demand. The outcomes demonstrate that the GTWR model outperforms the GWR model and exhibits significant statistical characteristics in their spatiotemporal distribution. Subsequently, based on the empirical results, the paper presents adjustment and improvement proposals for urban planning, design, management and policy in diverse areas of Xianyang, which holds great significance in promoting low-carbon travel and the sustainable development of Xianyang.

In summary, the frequency of bicycle travel can be augmented by enhancing public infrastructure, particularly by installing isolation facilities between motor vehicle lanes and non-motor vehicle lanes and strengthening the management of on-street parking. In established urban centers, it is crucial to concentrate on enhancing the convenience and safety of the cycling environment. In the emerging urban peripheries, infrastructure planning and improvement should center around strengthening land use diversity and the equalization of public service facilities. There exists a competitive relationship between bicycles and public buses. However, the construction of bicycle parking spaces can foster a cooperative relationship between the two travel modes.

The principal contributions of this study stem from two aspects. On the one hand, it discloses that the spatial heterogeneity of residents' cycling behavior is influenced not only by the quantity of facilities but also by the quality grades of facilities and business forms. Additionally, cycling safety facilities constitute another significant factor, thereby providing a more comprehensive perspective of the influencing factors for bicycle trips. On the other hand, the

research reveals that the influence of different facility qualities on cycling demand is mainly manifested in commuting trips and has a minor impact on flexible trips, which furnishes theoretical support for the formulation of management policies, such as staggered travel measures. This study also has certain limitations, microscale factors or environmental aesthetics, including greenery and trees, benches and streetlights, appealing buildings and landscapes should be taken into account in future research. Moreover, the sample size can be enlarged and groups can be segmented, which will assist in understanding the differences in bicycle travel among distinct groups and formulating feasible strategies and targeted approaches.

## Supporting information

**S1 Dataset. The dataset contains covering residents' cycling intention data, cycling trip information data, road network data, public service facility data, as well as other socio-economic data.** All these data are provided by the Xianyang Planning and Design Institute, which endows them with authority and reliability. Among them, the residents' cycling intention data originates from the residents' travel behavior survey carried out in the urban comprehensive transportation planning project commissioned by the Xianyang Municipal Government to the Xianyang Planning and Design Institute. The survey was comprehensively conducted in Xianyang from June 10th to 20th, 2022, and finally, 22,616 pieces of cycling data were successfully collected. During the data collection stage, we employed a strict and scientific sampling method, taking fully into account various characteristics of residents such as different ages, genders, occupations, and residential areas, striving to ensure that the sample has wide representativeness and can accurately reflect the actual situation of the residents in Xianyang. The survey content is meticulous and comprehensive, covering gender, age, income level, the ownership status of cars and bicycles, cycling departure time, the cycling departure location marked by latitude and longitude, cycling arrival time, as well as the corresponding latitude and longitude of the arrival location. It is particularly important to note that, in order to ensure the standardization of the research and the rights and interests of the participants, on the front page of the questionnaire, we elaborated on the research purpose in detail and obtained the written informed consent of the participants to participate in this academic research. For underage participants, they were only allowed to answer the questionnaire after obtaining the written informed consent of their parents or guardians. In terms of data quality control, we have adopted rigorous measures. We have carried out multiple rounds of data cleaning and verification on the collected data. Through strict data screening criteria, we have eliminated the obviously incorrect and abnormal data records, making every effort to ensure the accuracy and reliability of the data and providing a solid data foundation for the research. (CSV)

## Acknowledgments

The authors would like to thank Xianyang Planning and Design Institute and each member of the 905 research group for data collection and data processing. The authors also thank the respondents for providing data and information that are essential for this work.

## Author contributions

**Data curation:** Xiaohui Yan.

**Formal analysis:** Borui Yan.

**Investigation:** Xiaohui Yan, Borui Yan, Shuaiyang Jiao, Lei Zhang.

**Software:** Shuaiyang Jiao, Lei Zhang.

**Writing – original draft:** Xiaonan Zhang.

**Writing – review & editing:** Xiaonan Zhang.

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
