## [Decision Letter · Decision Letter 0]

3 Sep 2024

Dear Dr. Zhang,

Thank you for submitting your manuscript to PLOS ONE. After careful consideration, we feel that it has merit but does not fully meet PLOS ONE’s publication criteria as it currently stands. Therefore, we invite you to submit a revised version of the manuscript that addresses the points raised during the review process.

Reviewer 1 recommended reject while Reviewer 2 recommended major revision. They pointed out methodological concern and poor descriptions about the study efforts and achievements with many comments. I expect the authors to fully address the comments and appropriately upgrade the paper.

We look forward to receiving your revised manuscript.

Kind regards,

Hironori Kato, Dr. Eng.

Academic Editor

PLOS ONE

Journal Requirements:

2. You indicated that ethical approval was not necessary for your study. We understand that the framework for ethical oversight requirements for studies of this type may differ depending on the setting and we would appreciate some further clarification regarding your research. Could you please provide further details on why your study is exempt from the need for approval and confirmation from your institutional review board or research ethics committee (e.g., in the form of a letter or email correspondence) that ethics review was not necessary for this study? Please include a copy of the correspondence as an ""Other"" file.

3. We note that Figures 2, 3, 6 and 7 in your submission contain [map/satellite] images which may be copyrighted. All PLOS content is published under the Creative Commons Attribution License (CC BY 4.0), which means that the manuscript, images, and Supporting Information files will be freely available online, and any third party is permitted to access, download, copy, distribute, and use these materials in any way, even commercially, with proper attribution. For these reasons, we cannot publish previously copyrighted maps or satellite images created using proprietary data, such as Google software (Google Maps, Street View, and Earth). For more information, see our copyright guidelines: http://journals.plos.org/plosone/s/licenses-and-copyright.

a. You may seek permission from the original copyright holder of Figures 2, 3, 6 and 7 to publish the content specifically under the CC BY 4.0 license.  

Reviewers' comments:

Reviewer's Responses to Questions

**Comments to the Author**

1. Is the manuscript technically sound, and do the data support the conclusions?

Reviewer #1: No

Reviewer #2: Partly

2. Has the statistical analysis been performed appropriately and rigorously?

Reviewer #1: No

Reviewer #2: I Don't Know

3. Have the authors made all data underlying the findings in their manuscript fully available?

Reviewer #1: Yes

Reviewer #2: Yes

4. Is the manuscript presented in an intelligible fashion and written in standard English?

Reviewer #1: No

Reviewer #2: Yes

Reviewer #1: This study investigated the influencing factors on bicycle ridership in Xianyang, China. I have some severe concerns on this paper.

Major comments:

(1) The paper needs to be restructured in a more logical way. For example, in chapter 1, the GWR appears suddenly and the reader cannot follow the context, making it difficult to understand the contributions of the paper. Research questions and hypotheses should be clearly explained. Also, please explain the intention of each analysis. You need to explain logically whether the main contribution of this study is in the application of GWTR, in the data, in the application to Xianyang, or in the research questions.

(2) GWR is known to suffer from multicollinearity (see e.g., Murakami et al., spatial statistics 19, 68-89), and attention should be paid not only to global multicollinearity as analyzed in the paper, but also to location-specific local multicollinearity (e.g., by examining correlations between regression coefficient estimates or using diagnostic statistics for local multicollinearity). When local multicollinearity is strong, the regression coefficient estimates are almost impossible to interpret because they face identification problems. Also you should explain why you chose this method among various other spatially varying coefficients models (SVCM) that can mitigate local multicollinearity problem, such as multiscale GWR and Bayesian SVCM.

(3) Figures are not properly explained in the text, and there is no explanation of the vertical and horizontal axes. In the first place, all the figures are almost impossible to understand due to resolution problems. If Figure 5 is taken from ArcGIS, it should be indicated as such.

(4) GWR is a method for cross-sectional data, while GWTR is a method for time series cross-sectional (TSCS) or panel data. Then, why you can compare? How?

(5) The GWTR may be formulated with assuming that a sample is influenced by previous samples, but it is unclear how that part was formulated, weights were given, and bandwidth was estimated (or calibrated), since neither the method nor the results are described.

Minor comments:

(6) I could not understand the intention of the Moran I analysis of the explanatory variables in Table 5. Also, it is odd that sd is all 0.

(7) This paper needs to be native checked.

(8) It is not effective to distribute data in PDF format. Please consider to use other formats such as csv.

Reviewer #2: This paper explored the temporal and spatial distribution characteristic of cycling, identified the relationship between cycling behavior and built environments based on 2022 Daily Trip Survey in Xianyang, China. Then GTWR and GWR models are established include socio-economic factors, road factors and built environmental factors. In particular, four new findings are presented, mainly by setting two original indicators as built environmental factors. This also means that the findings are useful for a timely topic in a low- and middle-income country. However, there are some uncertainties and concerns regarding the characteristics of the data used, the statistical significance of the analysis results, and the validity of the discussion and conclusions, which require appropriate additions and corrections. In addition, the figure showing the spatial distribution of the regression coefficients is not very clear, and we strongly request that it be replaced with a clearer figure.

▼Page 8-9, Equations (1) to (3).

The formula needs to be laid out properly. In addition, the hat "�" seems to be garbled, so I ask for confirmation.

▼Page 8-12, Methodology.

Since previous studies comparing the results of the combined use of GWR and GTWR have been accumulated, including a paper by the developers of GTWR (Fotheringham and Yao, 2015 (https://doi.org/10.1111/gean.12071)), a review of the methodology should be included to clarify the position of this study. Therefore, it is necessary to clarify the position of this study by including a review of the methodology.

▼Page 15, Table 1.

The item name on the left, "Variables," is not a "Level 1 factor." Since the item names are different, you need to unify them into one of the two.

▼Page 16-17, Table 2.

Dependent Variable Y needs to be clarified specifically because it is unclear whether bicycle trips are counted on an origin basis or a destination basis.

▼Page 16, Line 320.

There is a space between "of" and "traffic."

▼Page 19, Table 4.

In each value, I wish to consider significant figures.

▼Figure 2.

As can be seen from Figure 2, there is some variation in the area of the research unit. Therefore, is there a possibility that there is a problem with the aggregation unit when setting the variables? It is necessary to clarify the appropriateness of the setting of the explanatory variables in terms of unit area.

▼Page 22-26, 5.3 Spatial characteristic of variables based on GTWR.

In order to evaluate the validity of the results in terms of statistical significance, isn't it necessary to show the spatial distribution of the t-values of the regression coefficients? This point needs to be explained.

In addition, to judge whether the statistical analysis was performed appropriately and rigorously, it is necessary to clarify what statistical application was used and, if there were any conditions set for the analysis.

▼Page 22, Line 375.

I think it should be "This" and not "his", so I would like it to be corrected.

▼Page 26-29, 5.4 Temporal heterogeneity.

When considering temporal heterogeneity, it is first necessary to clearly state how the t that appears in Equations 3 and 4 is set in the analysis, including its units.

▼Figures 3a and 3b.

The legends for Figures 3a and 3b need to be in English and the units should be shown.

▼Figures 6 and 7.

The graph showing the spatial distribution of regression coefficients has low visibility and we would like it to be replaced with something clearer.

The numerical values for each category in Figures 6 and 7 are difficult to read and I would like this to be improved.

**Do you want your identity to be public for this peer review?** For information about this choice, including consent withdrawal, please see our Privacy Policy

Reviewer #1: No

Reviewer #2: No

---

## [Author Response · Author response to Decision Letter 1]

2 Jan 2025

Dear Editors and Reviewers:

Thank you for your letter and for the reviewers’ comments concerning our manuscript entitled “Spatiotemporal Heterogeneity of Bicycle Ridership Based on GTWR Model

”(ID: PONE-D-24-25989). Those comments are all valuable and very helpful for revising and improving our paper, as well as the important guiding significance to our researches. We have studied comments carefully and have made correction which we hope meet with approval. The changes in the revised manuscript have been highlighted by red color. Point by point responses to the reviewers’ comments are listed below this letter.

Responds to the reviewer’s comments:

TO REVIEWER 1

General comments:

Reviewer #1: This study investigated the influencing factors on bicycle ridership in Xianyang, China. I have some severe concerns on this paper.

Major comments:

(1) The paper needs to be restructured in a more logical way. For example, in chapter 1, the GWR appears suddenly and the reader cannot follow the context, making it difficult to understand the contributions of the paper. Research questions and hypotheses should be clearly explained. Also, please explain the intention of each analysis. You need to explain logically whether the main contribution of this study is in the application of GTWR, in the data, in the application to Xianyang, or in the research questions.

(2) GWR is known to suffer from multicollinearity (see e.g., Murakami et al., spatial statistics 19, 68-89), and attention should be paid not only to global multicollinearity as analyzed in the paper, but also to location specific local multicollinearity (e.g., by examining correlations between regression coefficient estimates or using diagnostic statistics for local multicollinearity). When local multicollinearity is strong, the regression coefficient estimates are almost impossible to interpret because they face identification problems. Also you should explain why you chose this method among various other spatially varying coefficients models (SVCM) that can mitigate local multicollinearity problem, such as multiscale GWR and Bayesian SVCM.

(3) Figures are not properly explained in the text, and there is no explanation of the vertical and horizontal axes. In the first place, all the figures are almost impossible to understand due to resolution problems. If Figure 5 is taken from ArcGIS, it should be indicated as such.

(4) GWR is a method for cross-sectional data, while GTWR is a method for time series cross-sectional (TSCS) or panel data. Then, why you can compare? How?

(5) The GTWR may be formulated with assuming that a sample is influenced by previous samples, but it is unclear how that part was formulated, weights were given, and bandwidth was estimated (or calibrated), since neither the method nor the results are described.

Minor comments:

(6) I could not understand the intention of the Moran I analysis of the explanatory variables in Table 5. Also, it is odd that sd is all 0.

(7) This paper needs to be native checked.

(8) It is not effective to distribute data in PDF format. Please consider to use other formats such as csv.

Response:

Thank you very much for the detailed comments. We appreciate your time and help in reviewing our manuscript, and the insightful comments you provided that have helped significantly improve the quality of this study. We have revised the paper very carefully according to your suggestions, and detailed explanations of all the issues are as follows.

The paper needs to be restructured in a more logical way. For example, in chapter 1, the GWR appears suddenly and the reader cannot follow the context, making it difficult to understand the contributions of the paper. Research questions and hypotheses should be clearly explained. Also, please explain the intention of each analysis. You need to explain logically whether the main contribution of this study is in the application of GTWR, in the data, in the application to Xianyang, or in the research questions.

Response:

Thank you very much for the Reviewer's thorough review of the paper and their valuable feedback. We fully agree with the Reviewer's identification of the structural issues in the paper. Based on the expert's suggestions, we have revisited and adjusted the structure of the paper to ensure logical coherence from the research background to the research questions, hypotheses, methods, and results, thereby enhancing its readability. The specific adjustments include:

Enhancing the background introduction: Expanding the background content in Chapter 1 to provide necessary background information for the introduction of GWR, ensuring that readers can smoothly understand its role in the research.

Clarifying research questions and hypotheses: Revisiting and clearly stating the research questions and hypotheses to ensure that they are closely linked to the research objectives and methods.

Elucidating the intentions of the analyses: Providing detailed explanations of the purpose and significance of each analysis in the methods section, so that readers can understand how they support the research objectives.

Highlighting the research contributions: Clearly pointing out the main contribution points of this study, whether they lie in the application of GTWR, the data, the specific application to Xianyang, or the research questions themselves.

GWR is known to suffer from multicollinearity (see e.g., Murakami et al., spatial statistics 19, 68-89), and attention should be paid not only to global multicollinearity as analyzed in the paper, but also to location specific local multicollinearity (e.g., by examining correlations between regression coefficient estimates or using diagnostic statistics for local multicollinearity). When local multicollinearity is strong, the regression coefficient estimates are almost impossible to interpret because they face identification problems. Also you should explain why you chose this method among various other spatially varying coefficients models (SVCM) that can mitigate local multicollinearity problem, such as multiscale GWR and Bayesian SVCM.

Response:

Many thanks for your question.

We fully understand and agree with the issue you have pointed out, namely that Geographically Weighted Regression (GWR) may be affected by multicollinearity, and this impact is not limited to the global scope but may also manifest in specific locations. Under your guidance, we have revisited the analysis process in our paper and paid special attention to the issue of local multicollinearity. To more comprehensively assess the impact of multicollinearity on GWR results, we have added a Local Spatial Autocorrelation Test in the revised manuscript to detect the spatial autocorrelation of travel frequency data, which helps us understand the spatial distribution characteristics of the data and potential collinearity issues.

We are deeply grateful for the invaluable feedback provided by the reviewer. Indeed, MGWR, as an improved geographical weighted regression method, enhances the performance of the GWR model by assigning a distinct bandwidth to each predictor variable. However, in our study, we focus on the spatiotemporal heterogeneity of the impact of the built environment on travel behavior. While MGWR excels in analyzing spatial heterogeneity, it unfortunately does not account for the uneven temporal distribution of sample points, which limits its applicability in our research context.

Similarly, the SVCM model, another powerful spatial varying coefficient model, also fails to directly describe the temporal heterogeneity of sample points. Our research aims to delve into the impact of the built environment on travel behavior from both spatial and temporal dimensions, making it particularly important to select a model that can capture both spatiotemporal heterogeneities.

Once again, we express our sincere thanks to the reviewer for their precious suggestions. In future studies, we will further explore the applications of models such as MGWR and SVCM, aiming to more comprehensively reveal the complex relationship between the built environment and travel behavior. At the same time, we will continue to keep abreast of new developments in this field and continuously optimize and improve our research methods.

The modified contents have been marked in red font on Page 22 line 391-397 and Page 23 line 398-407 in the updated manuscript.

Figures are not properly explained in the text, and there is no explanation of the vertical and horizontal axes. In the first place, all the figures are almost impossible to understand due to resolution problems. If Figure 5 is taken from ArcGIS, it should be indicated as such.

Response:

We are extremely grateful to the reviewer for the invaluable feedback regarding the clarity of the figures and tables in our paper.

We fully recognize the significance of figures and tables in scientific research, as well as the pivotal role that clear and detailed figure/table explanations play in facilitating readers' understanding of research content and conclusions. In response to the reviewer's concerns about insufficient explanations and low resolution, we have taken the following measures to improve:

(1) Enhancing Figure and Table Resolution

We sincerely apologize for the lack of clarity in Figures 6 and 7(Figure 6 and Figure 7 in the original manuscript correspond to Figure 9 and Figure 10 in the revised manuscript respectively), particularly in displaying the spatial distribution of regression coefficients and categorical values. To elevate the quality of our paper, we have re-generated all figures in the revised manuscript, ensuring they possess adequate resolution and optimized font sizes and color contrasts for readability across all categorical values.

(2) Providing Detailed Explanations for Figures and Tables

For each figure and table, we have incorporated detailed explanatory text in the revised manuscript, explicitly elucidating their meanings. For vertical and horizontal axes, we have clearly labeled their significance, units, and ranges to facilitate comprehension.

(3) Pre-Submission Self-Check

Following these revisions, we have conducted a meticulous self-review and proofreading to ensure that all figures and tables comply with the publication requirements of SCI journals.

GWR is a method for cross-sectional data, while GTWR is a method for time series cross-sectional (TSCS) or panel data. Then, why you can compare? How?

Response:

Many thanks for your question.

The GWR model is applicable to cross-sectional data, while the GTWR model is suitable for panel data. The survey data in this study covers a period of three days, with each participant required to fill in information based on a single day or trip. Given the short timeframe of three days and no changes in the built environment, all travelers' trip behaviors within these three days can be considered as occurring at a single time point, making the trip data eligible for cross-sectional analysis and, thus, applicable to the GWR model. Additionally, travelers' trips inherently possess a temporal attribute, with each trip occurring at a different time point within a 24-hour day. Therefore, when examining all travelers' trips across the day, a clear temporal characteristic emerges. By linking trips with their respective time attributes, the data can be viewed as panel data, suitable for the GTWR model.

Hence, there are subtle differences in applying the two models to the same dataset.

According to extensive literature, the GTWR model, due to its consideration of the temporal dimension, generally outperforms the GWR model in terms of model fit, residuals, and other metrics. In practical applications, if the data is purely cross-sectional, only the GWR model can be used. If the data is panel data, the GTWR model can be applied, or the data can be processed into cross-sectional data (e.g., through time aggregation or averaging) before applying the GWR model. However, losing the original temporal attributes of the data may lead to a decrease in model accuracy.

In terms of theoretical foundations, the GWR model does not consider the temporal attribute of sample points, assuming all sample points occur at the same time. In contrast, the GTWR model's sample points are associated with time attributes, allowing for the analysis of heterogeneity in the temporal distribution of sample points and the uneven impact of influencing factors on the dependent variable over time.

The GTWR may be formulated with assuming that a sample is influenced by previous samples, but it is unclear how that part was formulated, weights were given, and bandwidth was estimated (or calibrated), since neither the method nor the results are described.

Response:

We are deeply grateful to the reviewer for their thorough attention and valuable feedback on the details of the GTWR method's construction.

We acknowledge that the specific construction process, weight assignment, and bandwidth estimation methods of the GTWR method were not sufficiently detailed in the paper, which may have posed difficulties for readers in understanding this section. In the revised manuscript, we have provided a more detailed description of the GTWR construction process. Specifically:

Clarify the construction process: We will elaborate on the basic assumptions, mathematical model, and construction steps of the GTWR method, enabling readers to clearly understand how GTWR is constructed based on the interdependence between samples.

Explain weight assignment: We will explain how weights are assigned in the GTWR method based on spatial relationships, temporal sequences, or other relevant factors between samples, and the role of these weights in model construction and result interpretation.

Describe bandwidth estimation methods: We will introduce in detail the principles, methods, and specific implementation steps of bandwidth estimation, including how to select appropriate bandwidth estimation methods based on data characteristics and research objectives, and the impact of bandwidth estimation results on model performance.

Additionally, we will supplement the corresponding results section, demonstrating the effectiveness of the GTWR method in practical applications through examples, as well as the specific impact of weight assignment and bandwidth estimation on model outcomes.

The modified contents have been marked in red font on Page 11 line 222-223 and Page 12 line 245-246 in the updated manuscript.

I could not understand the intention of the Moran I analysis of the explanatory variables in Table 5. Also, it is odd that sd is all 0.

Response:

Special thanks to you for reviewing our paper and providing valuable feedback.

Regarding the Moran's I analysis of the explanatory variables in Table 5 that you mentioned, our intention was to assess the spatial autocorrelation of the explanatory variables through this analysis, thereby understanding whether they exhibit trends of spatial clustering or dispersion. A Moran's I index greater than 0 indicates spatial positive correlation, and the larger the value, the stronger the spatial correlation. Conversely, a Moran's I index less than 0 suggests spatial negative correlation, and the smaller the value, the greater the spatial variation. When Moran's I equals 0, the spatial distribution is random. The significance of the hypothesis test is measured by the p-value. When p < 0.05, the null hypothesis (that the explanatory variables are spatially random) is rejected, indicating the presence of spatial correlation among the explanatory variables, which meets the condition for Geographically Weighted Regression (GWR) modeling. A z-score close to 0 suggests no significant clustering within the study area. A positive z-score indicates clustering of high values, while a negative z-score indicates clustering of low values. By combining the results of the z-score and Moran's I, if both are positive (or negative) and the z-score is large (or small), it is considered that there is significant spatial clustering for that variable. Therefore, in this paper, we first calculated the Moran's I index for each explanatory variable, determined the presence of spatial correlation among the variables based on the p-va

---

## [Decision Letter · Decision Letter 1]

16 Feb 2025

Spatiotemporal Heterogeneity of Bicycle Ridership Based on GTWR Model

PONE-D-24-25989R1

Dear Dr. Zhang,

We’re pleased to inform you that your manuscript has been judged scientifically suitable for publication and will be formally accepted for publication once it meets all outstanding technical requirements.

Kind regards,

Hironori Kato, Dr. Eng.

Academic Editor

PLOS ONE

Editor Comments:

Both reviewers recommended accepted. Congratulations for successful revisions!

Reviewers' comments:

Reviewer's Responses to Questions

**Comments to the Author**

Reviewer #1: All comments have been addressed

Reviewer #2: All comments have been addressed

2. Is the manuscript technically sound, and do the data support the conclusions?

Reviewer #1: Partly

Reviewer #2: Yes

3. Has the statistical analysis been performed appropriately and rigorously?

Reviewer #1: Yes

Reviewer #2: Yes

4. Have the authors made all data underlying the findings in their manuscript fully available?

Reviewer #1: Yes

Reviewer #2: Yes

5. Is the manuscript presented in an intelligible fashion and written in standard English?

Reviewer #1: Yes

Reviewer #2: Yes

Reviewer #1: I believe this paper has been substantially revised. No further comment from me.

I believe this paper has been substantially revised. No further comment from me.

Reviewer #2: I have confirmed that your response to the peer review has been sufficiently addressed. Therefore, I have no additional comments.

**Do you want your identity to be public for this peer review?** For information about this choice, including consent withdrawal, please see our Privacy Policy

Reviewer #1: No

Reviewer #2: **Yes: ** Tetsuharu Oba

---

## [Editor Report · Acceptance letter]

PONE-D-24-25989R1

PLOS ONE

Dear Dr. Zhang,

I'm pleased to inform you that your manuscript has been deemed suitable for publication in PLOS ONE. Congratulations! Your manuscript is now being handed over to our production team.

Kind regards,

on behalf of

Dr. Hironori Kato

Academic Editor

PLOS ONE